# A common pathway controls cell size in the sepal and leaf epidermis leading to a nonrandom pattern of giant cells

Frances K. Clark[1,2☉], Gauthier Weissbart[3☉], Xihang Wang[2☉], Kate Harline[2], Chun-Biu Li[4], Pau Formosa-Jordan[3,5,6]*, Adrienne H. K. Roeder[2,6]*

1 Department of Molecular Biology and Genetics, Cornell University, Ithaca, New York, United States of America, 2 Weill Institute for Cell and Molecular Biology and School of Integrative Plant Science, Section of Plant Biology, Cornell University, Ithaca, New York, United States of America, 3 Department of Plant Developmental Biology, Max Planck Institute for Plant Breeding Research, Cologne, Germany, 4 Department of Mathematics, Stockholm University, Stockholm, Sweden, 5 Cluster of Excellence on Plant Sciences (CEPLAS), Max Planck Institute for Plant Breeding Research, Cologne, Germany, 6 Polyploidy Integration and Innovation Institute

☉ These authors contributed equally to this work.
* pformosa@mpipz.mpg.de (PF-J); ahr75@cornell.edu (AHKR)

## Abstract

Arabidopsis leaf epidermal cells have a wide range of sizes and ploidies, but the mechanisms patterning their size and spatial distribution remain unclear. Here, we show that the same genetic pathway creating giant cells in sepals also regulates cell size in the leaf epidermis, leading to the formation of giant cells. In both sepals and leaves, giant cells are scattered among smaller cells; therefore, we asked whether their spatial arrangement is random. By comparing sepal and leaf epidermises with computationally generated randomized tissues, we show that the giant cell pattern becomes less random across the epidermis as the cells surrounding giant cells divide, leading to clustered patterns in mature tissues. Our cell-autonomous and stochastic computational model reproduces the giant cell organization, suggesting that random giant cell initiation together with the divisions of surrounding cells lead to the observed clustered pattern. These findings reveal that cell-size patterning is developmentally regulated by common mechanisms in leaves and sepals, and the spatial pattern of giant cells emerges from the interplay between stochastic cell-autonomous gene expression and tissue growth.

## Introduction

The *Arabidopsis thaliana* (hereafter Arabidopsis) mature leaf blade epidermis contains three main cell types: stomatal guard cells, trichomes, and pavement cells [1]. Stomatal guard cells surround stomatal pores through which gas exchange occurs, and trichomes are large branched hair cells that serve to discourage herbivory, among other functions

**Data availability statement:** Microscopy data, MorphoGraphX meshes for the images, as well as code and material used for data analysis and modeling are available at Open Science Framework (osf.io), DOI: https://doi.org/10.17605/OSF.IO/RFCWS.

**Funding:** This work was funded by the National Science Foundation (US NSF, https://www.nsf.gov) Integrative Organismal Systems IOS-1553030 (AHKR), Division of Biological Infrastructure DBI-232051 (AHKR and PF-J); the International Max Planck Research School (https://www.mpipz.mpg.de/imprs/) on "Understanding Complex Plant Traits Using Computational and Evolutionary Approaches" (GW); a core grant from the Max Planck Society (https://www.mpg.de/) (PF-J); and the Deutsche Forschungsgemeinschaft (DFG, German Research Foundation, https://www.dfg.de) under Germany's Excellence Strategy—EXC-2048/1—project ID 390686111 (PF-J). This research was advanced through time spent collaborating at the KITP program on Dynamics of Self-Organization in Animal and Plant Development, supported in part by the National Science Foundation (https://www.nsf.gov) Division of Physics under Grant No. PHY-1748958 and Gordon and Betty Moore Foundation (https://www.moore.org) Grant No. 2919.02. The funders did not play any role in the study design, data collection and analysis, decision to publish, or preparation of the manuscript.

**Competing interests:** The authors have declared that no competing interests exist.

**Abbreviations:** ABRC, Arabidopsis Biological Resource Center; ACR4, ARABIDOPSIS CRINKLY 4; ATML1, *Arabidopsis thaliana* MERISTEM LAYER1; DEK1, DEFECTIVE KERNEL; dpg, days postgermination; LGO, LOSS OF GIANT CELLS FROM ORGANS; SMR1, SIAMESE-RELATED 1; TFP, teal fluorescent.

[2]. All other epidermal cells in the mature leaf blade epidermis (the expanded part of the leaf between the midrib and the margin) are classified as pavement cells. However, pavement cells are not a homogeneous group of cells, but rather exhibit a variety of sizes, ploidies, and shapes [3,4]. Much research has focused on the patterning of stomata [5–7] and trichomes [8,9], leading to important insights into how the regulation of intercellular signaling, cell fate specification, the cell cycle, and polarized cell division orientation give rise to their spatial arrangement. However, the patterning of pavement cells is understudied. In particular, little is known about how some pavement cells are specified to become larger and more highly polyploid than others.

Pavement cell-size patterning has been studied in the Arabidopsis sepal. Pavement cells in the sepal vary in size and ploidy, with some cells reaching up to 800 μm in length (Fig 1A) and having ploidies up to 32C [10]. These very large pavement cells that have a characteristic highly anisotropic shape and bulge out of the epidermis have been named "giant cells" [10], and these form when a cell endoreduplicates early during growth [10]. Endoreduplicating cells replicate their DNA but do not enter mitosis or divide and instead continue to grow and increase their ploidy. Once a cell enters endoreduplication, it is thought to terminally differentiate and almost never re-enters the mitotic cycle [10]. Similar numbers of giant cells form on sepals within an Arabidopsis plant and among plants, but the precise spatial arrangement of giant cells differs from sepal to sepal.

Forward-genetic screens have identified the genes involved in sepal giant cell patterning, and double mutant analysis has allowed these genes to be ordered within a genetic pathway [10–13] (Fig 1). The homeodomain leucine zipper Class IV transcription factor *Arabidopsis thaliana* MERISTEM LAYER1 (ATML1) promotes giant cell specification in a dose-dependent manner [11,12]. Loss of *ATML1* function in sepals greatly reduces giant cell number, and overexpression of *ATML1* leads to ectopic giant cell formation (Fig 1A, 1C, and 1G) [11,12]. ATML1 protein concentration fluctuates in the protodermal nuclei of developing sepals [12]. High concentrations of ATML1 reached during the G2 phase of the cell cycle are strongly correlated with giant cell differentiation, consistent with a model in which an ATML1 concentration that surpasses a threshold in G2 results in giant cell specification, early endoreduplication, and giant cell differentiation [12]. The receptor-like kinase ARABIDOPSIS CRINKLY 4 (ACR4) functions upstream of ATML1 to promote giant cell formation [11,12,14–16] (Fig 1B and 1H). Loss of function of *ACR4* leads to a modest reduction in the number of giant cells [11] (Fig 1A and 1B). The calpain protease DEFECTIVE KERNEL (DEK1) and the CDK inhibitor LOSS OF GIANT CELLS FROM ORGANS (LGO; also known as SIAMESE-RELATED 1, SMR1) function genetically downstream of ATML1 to promote giant cell formation [12] (Fig 1H). A hypomorphic mutant *dek1* allele *(dek1-4)* results in the complete loss of giant cells from sepals [11] (Fig 1D). Similarly, sepals from plants homozygous for a loss-of-function mutation in *LGO* have no giant cells [10,11] (Fig 1E), and overexpression of *LGO* increases giant cell number [11] (Fig 1F). It is unknown whether this genetic pathway affects cell size only in the sepal or whether it is also a more general mechanism of epidermal cell-size patterning in other organs.

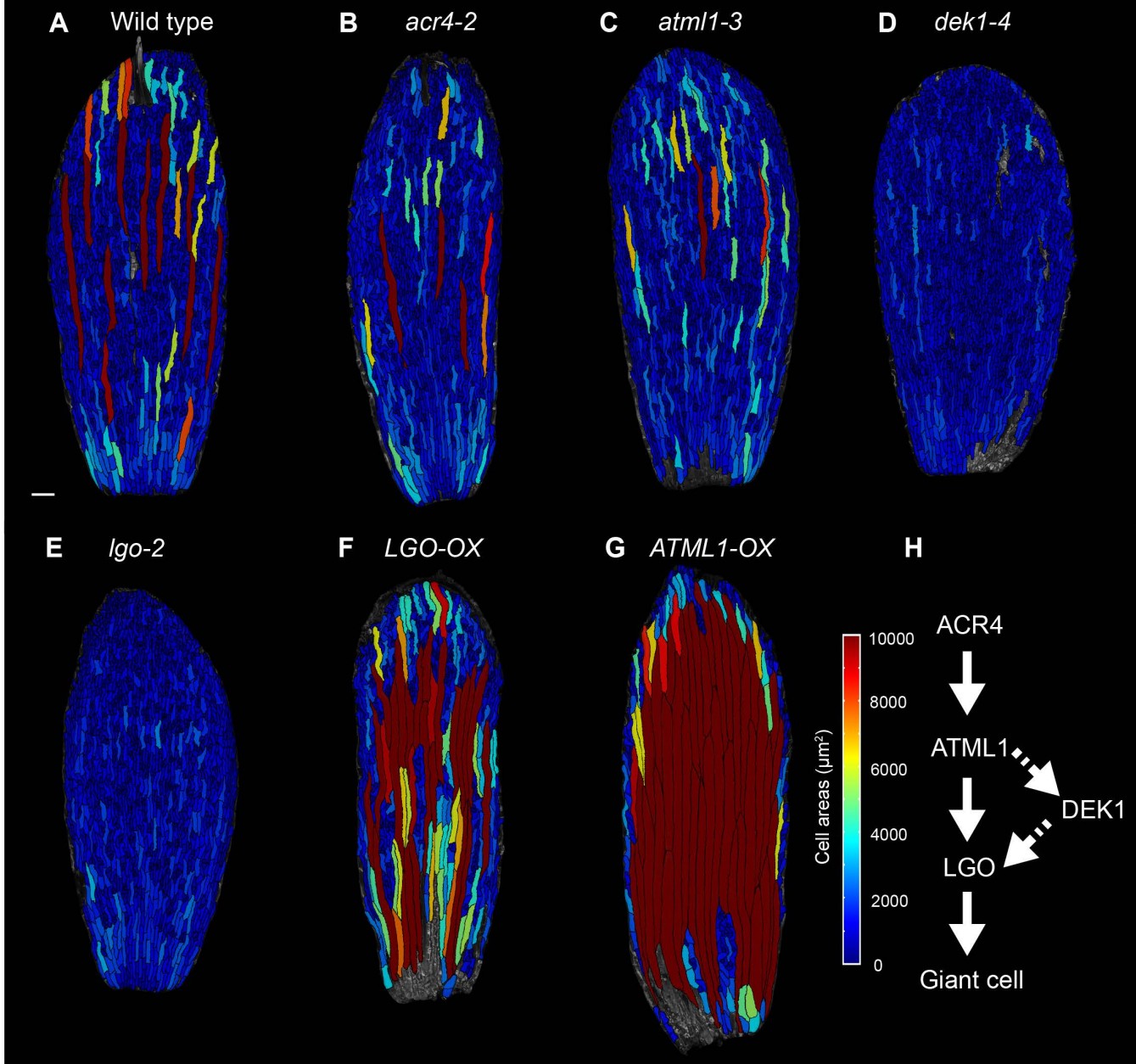

**Fig 1. The genetic pathway that regulates giant cell development in sepals. (A–G)** Cell area heat maps (in µm²) of the abaxial (outer) surface of a stage 14 adult sepal of **(A)** wild type, **(B)** *acr4-2*, **(C)** *atml1-3*, **(D)** *dek1-4*, **(E)** *lgo-2*, **(F)** *LGO-OX* (*pATML1::LGO*), and **(G)** *ATML1-OX* (*pPDF1::FLAG-ATML1*). Scale bar represents 100 µm. **(H)** The ordering of genes into a genetic pathway according to double-mutant phenotypic analysis. The underlying data for this figure can be found at Open Science Framework (osf.io), https://doi.org/10.17605/OSF.IO/RFCWS.

Leaf pavement cell size is affected by the family of CDK inhibitors that includes LGO, known as the SIAMESE/SIAMESE-RELATED (SIM/SMR) family [17,18]. SMR proteins bind to cyclin CDK complexes and inhibit their phosphorylation of downstream targets [18]. *lgo* mutants lack large pavement cells and have a reduction in leaf cell endoreduplication compared with wild type [10,17,18]. In *lgo* mutants, pavement cells that should be mature continue to divide [19]. Furthermore, overexpression of the closely

related paralog of *LGO*, *SIM*, results in larger and more highly endoreduplicated leaf epidermal pavement cells [17]. In sepals, LGO upregulates defense response genes, including glucosinolate biosynthesis genes [20], whereas in leaves, ATML1 promotes the formation of ER bodies, which contain components of the glucosinolate system, in large pavement cells [21], suggesting a common role for large cells in defense response. Whether the same upstream components of the sepal giant cell pathway also function in leaf cell-size patterning has not been thoroughly investigated. One study compared pavement cell size in *dek1-4* and wild-type cotyledons and found no evidence that the cells differed in ploidy [22]. However, true leaves were not examined.

In leaves and sepals, it is unknown whether giant cells exhibit a spatially ordered pattern across the organ, or if instead their spatial arrangement is random. Other epidermal cell types are nonrandomly distributed across the leaf tissue. For instance, trichomes do not form in adjacent cells due to lateral inhibition mediated by diffusible signals [9], and stomata rarely differentiate in adjacent cells due to both lineage-specific division orientation and intercellular signaling [23]. In contrast to stomata and trichomes, sepal giant cells can be in contact with one another. However, it is unknown whether such giant cell contacts are likely to be formed by chance. Due to their large shapes, quantifying the spatial arrangement of giant cells has remained challenging, and standard methods for assessing point pattern randomness are not applicable [24–27].

Here, we imaged and analyzed large areas of leaves to obtain a holistic understanding of both the cell-size distributions and the spatial arrangements of epidermal pavement cells in the leaf blade. We discovered that the genetic pathway that controls sepal giant cell formation also has a broader role in patterning epidermal pavement cell size in leaves. We quantified the spatial organization of giant cells using simulated randomized tissues and found that giant cells tend to cluster together in both mature leaves and sepals more than expected by chance, reflecting the tissue's developmental history. Using modeling and data analysis, we found that giant cells emerge randomly in space at early stages of development, but the division of surrounding cells causes the spatial pattern to become nonrandom and clustered within the context of the whole tissue over time. Our computational modeling supports the notion that a nonrandom clustered pattern can emerge in a proliferating tissue over developmental time in a cell-autonomous and stochastic manner.

## Results

### Arabidopsis leaves exhibit a large range of pavement cell sizes, similar to sepals

In sepals, giant cells are easily visible because they are highly elongated (S1A Fig). Similarly, we observe large and highly anisotropic cells in cauline leaves (S1B Fig). In rosette leaves, pavement cells of the epidermis are jigsaw puzzle-piece shaped with lobes and necks, such that cell size is not readily apparent by eye (S1C Fig); however, heterogeneity in pavement cell sizes has been previously observed [4,28]. Therefore, we wondered to what extent the distribution of cell sizes observed in sepals, ranging from giant cells to small cells, also occurs in rosette leaves. We imaged large sections of the blade (excluding midrib and margin cells) of leaf 1 or 2 from wild-type plants at 25 days postgermination (dpg) when the leaves are fully expanded and mature. Leaves 1 and 2 initiate simultaneously and are indistinguishable; therefore, we refer to them interchangeably as leaf 1 or 2. We measured the area of the epidermal cells of leaves 1 or 2 and sepals on the abaxial (bottom) side (Fig 2A and 2C). Throughout our analysis, we use the term size to mean cell area because area has been shown to be a more relevant measure of cell size than volume in highly vacuolated plant epidermal cells [4]. We observed that the abaxial cell-size distributions for both sepals and leaves are asymmetric, with long tails representing large cells (Fig 2E). Sepal giant cells are larger outliers in size and, consequently, the cell-size distribution in the sepal has a more extended tail than in the leaf (Fig 2E). Still, we observed that the cell-size range in the leaf and sepal are similar and the largest cells of the sepal are about the same size as the largest cells of the leaf (Fig 2A–2E). We conclude that Arabidopsis leaves have a diverse range of cell sizes characterized by a long-tailed distribution, similar to the abaxial side of sepals.

### Large cells are formed on the adaxial side as well as the abaxial side of the leaf

In sepals, giant cells are restricted to the abaxial (outer) surface (Figs 2A, 2B, and S2A–S2D). We asked whether there was a difference in cell size between adaxial (top) and abaxial (bottom) surfaces of the leaf. Large cells of similar size

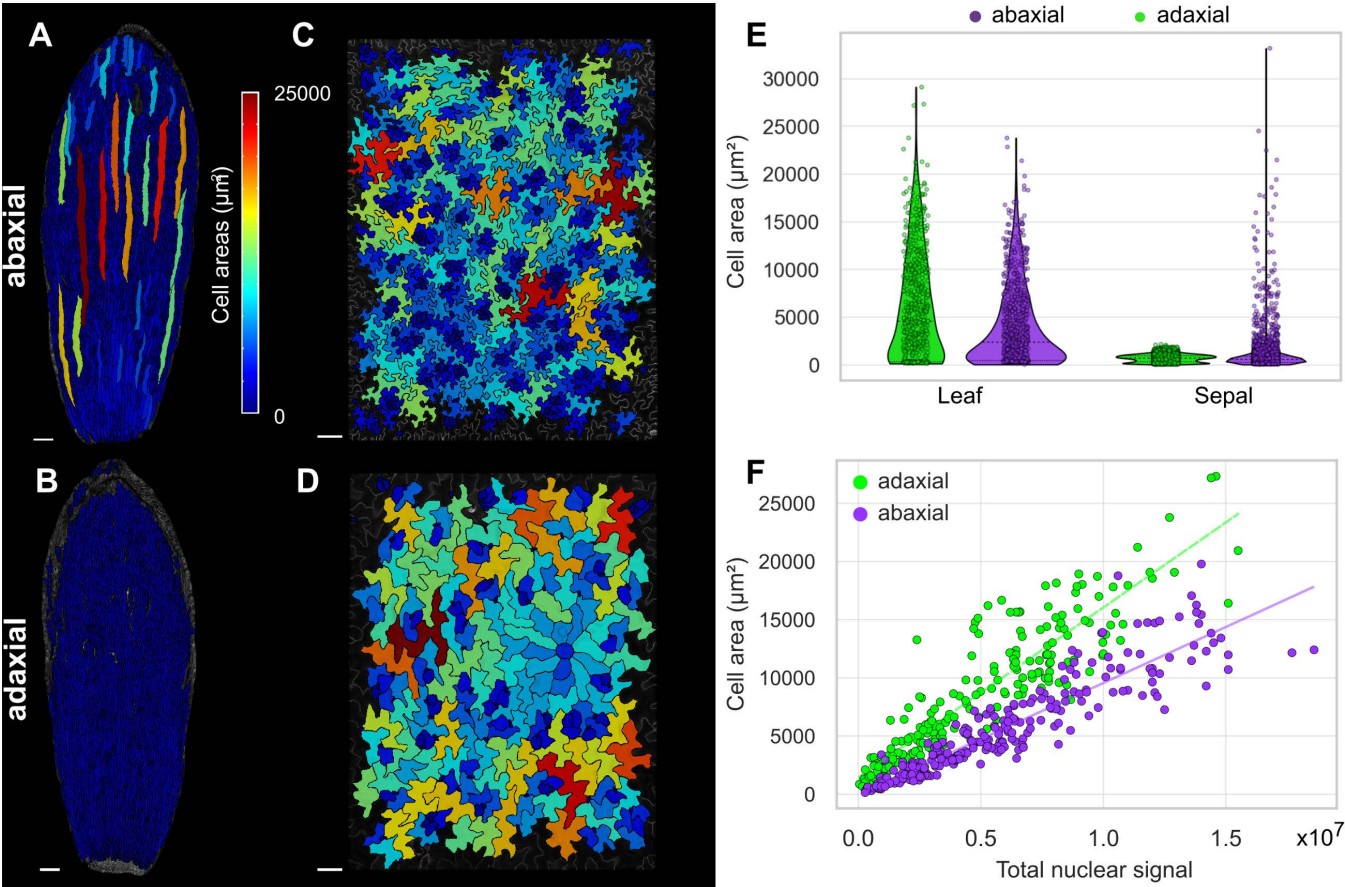

**Fig 2. Abaxial and adaxial cell-size distribution in the wild-type leaf and sepal epidermis; size correlates with DNA content. (A–D)** Cell area heat maps (in μm²) of (**A**) abaxial surface of wild-type sepal, (**B**) adaxial surface of wild-type sepal, (**C**) abaxial surface of 25-dpg wild-type leaf 1 or 2 (cell density: 234 cells mm⁻²), and (**D**) adaxial surface of 25-dpg wild-type leaf 1 or 2 (cell density: 156 cells mm⁻²). Scale bars represent 100 μm. (**E**) Violin and strip plots of cell areas of abaxial and adaxial sides of 25-dpg wild-type leaves (two pooled replicates) and adult wild-type sepals (three pooled replicates). (**F**) Adaxial side (green) and abaxial side (purple) of 25-dpg leaf cell area versus DNA content as measured by H2B-TFP total nuclear fluorescence, with $R^2 = 0.85$ for the abaxial side and $R^2 = 0.82$ for the adaxial side (one of two replicates). See S2 Fig for replicates. The underlying data for this figure can be found at Open Science Framework (osf.io), https://doi.org/10.17605/OSF.IO/RFCWS.

are formed on both the adaxial and abaxial surfaces of the leaf, in contrast to the sepal (Figs 2A–2E and S2A–S2F). In addition, across the whole cell-size distribution, the cell density is lower and many cells are slightly more expanded on the adaxial side than the abaxial side (Figs 2C, 2D, S2E, and S2F). There are a greater number of stomata and stomatal lineage cells (very small cells) on the abaxial side compared with the adaxial side (Figs 2C–2E, S2E, and S2F). We also observed that the abaxial cells are more lobed than the adaxial cells (Figs 2C, 2D, S2E, and S2F). Despite slight differences, the cell-size distributions of the abaxial and adaxial sides of the leaf are quite similar, particularly in the tails, where both sides exhibit a similar range of larger cells, in contrast to the sepal, where only the abaxial side has very large cells.

## Cell area correlates with DNA content

Cell area and ploidy are positively correlated in leaf epidermal cells [4]. A recent study showed that fluorescence levels of histone fluorescence reporters measured from microscopy images are a good proxy to infer nuclear ploidy in Arabidopsis cotyledons [29]. Hence, to validate whether cell area and ploidy were correlated in our leaves, we measured DNA content

by quantifying total fluorescence of Histone 2B-TFP (*pUBQ10::H2B-TFP*) within each cell nucleus of the 25-dpg leaf images. For this reporter, the H2B-TFP signal distribution was noisy and continuous, not divided into four discrete ploidy peaks, providing an approximation of ploidy, not exact DNA content. As expected, a strong linear correlation between H2B-TFP fluorescence and cell area was observed for both the abaxial pavement cells ($R^2 = 0.85$ and 0.91; $n = 2$) and the adaxial pavement cells ($R^2 = 0.79$ and 0.82; $n = 2$) (Figs 2F and S2G). Therefore, we focus on analyzing cell size, and infer that large cell size indicates high ploidy.

We wondered whether cells of similar size on the abaxial and adaxial side of the same leaf also have a similar DNA content. We found that cells of similar DNA content are larger on the adaxial side than on the abaxial side (Figs 2F and S2G), suggesting that adaxial cells have expanded more than abaxial cells, as noted above. Because the largest sepal cells and the largest leaf cells had approximately the same areas, we asked whether the DNA content of these cells was also similar. We found that the total H2B-TFP fluorescence values of the largest cells were very similar between sepal and leaf, suggesting that these largest cells are similar in ploidy (S2H Fig).

### Cell-size patterning emerges at the tip and progresses basipetally as the leaf differentiates

To determine how the cell-size pattern emerges in the leaf during development, we imaged both the adaxial and abaxial surfaces of each leaf at different stages of development. From 5 to 9 dpg, cell size increases greatly (S3 Fig), as expected. At day 5, cells throughout the blade are fairly homogeneous in size, with a few cells starting to expand near the distal tip, and the large cells of the margin and overlying midrib already apparent (Fig 3A). We focus on pavement cells in the blade and exclude margin and midrib cells from further analysis. The cell-size pattern consisting of large cells interspersed between small cells progressively develops basipetally from the tip (Fig 3A–3C), whereas at the base the cells remain uniformly small. The progression of cell-size patterning down the leaf is consistent with the well-established basipetal wavefront of differentiation and cessation of cell division [30]. The cell area distributions showed that more large cells appear throughout development and the maximal cell size increases (Fig 3A, 3B, and 3D). By 9 dpg, cell size has been patterned almost to the base of the leaf (Fig 3A and 3B).

We next asked whether the wavefront of cell-size patterning progresses basipetally at the same rate on the abaxial and adaxial sides of the leaf. Using images of both the abaxial and adaxial sides of the same leaf, we plotted the positions of the centers of the largest cells on both sides (including margin and midrib cells for landmarks) (Fig 3C). Large pavement cells are observed in the same proximal–distal region on abaxial and adaxial sides during development (Fig 3C). The region expands in the proximal direction as development progresses. This suggests that the wavefront of patterning and differentiation is coordinated across the abaxial/adaxial axis of the leaf.

### The sepal giant cell specification pathway also patterns giant cells in leaves

Because the cell-size distributions have similarities in leaves and sepals, we tested whether the giant cell specification pathway in sepals (Fig 1H) also functions in the leaf to pattern cell size. We imaged leaf 1 or 2 at both 9 and 25 dpg from wild-type and giant cell pathway mutants. At 9 dpg, patterning has just extended to the base of the leaf, and the leaf is still small enough that we could image the whole upper abaxial quadrant to determine the pattern over a large fraction of the leaf blade (Figs 4A, S4, and S5). At 25 dpg, the leaf is fully differentiated, fully expanded, and the pattern is established (Figs 4B, S6, and S7). We found that cell-size patterning in the leaf is similarly affected in the mutants at both 9 and 25 dpg as in the mature sepal (Figs 4, S8A, and S8B). Notably, the largest cells show similar variations in their numbers across genotypes. Similar to the sepal, the size of the largest cells is moderately reduced in *acr4-2* mutants (Figs 1B and 4A–4D), and more greatly reduced in *atml1-3* mutants (Figs 1C and 4A–4D). The reduction in large cells is drastic in *dek1-4* and *lgo-2* mutant sepals and leaves, resulting in the absence of a long tail in the cell-size distribution (Figs 1D–1E and 4A–4D). For these genotypes, the number of medium-sized cells is also substantially decreased (Figs 1D, 1E, 4C, and 4D). Conversely, the overexpression of *ATML1* (*ATML1-OX*) or *LGO* (*LGO-OX*) leads to an increase in the size of large cells and in fewer small cells compared with wild type, as in the sepal (Figs 1F, 1G, and 4A–4D).

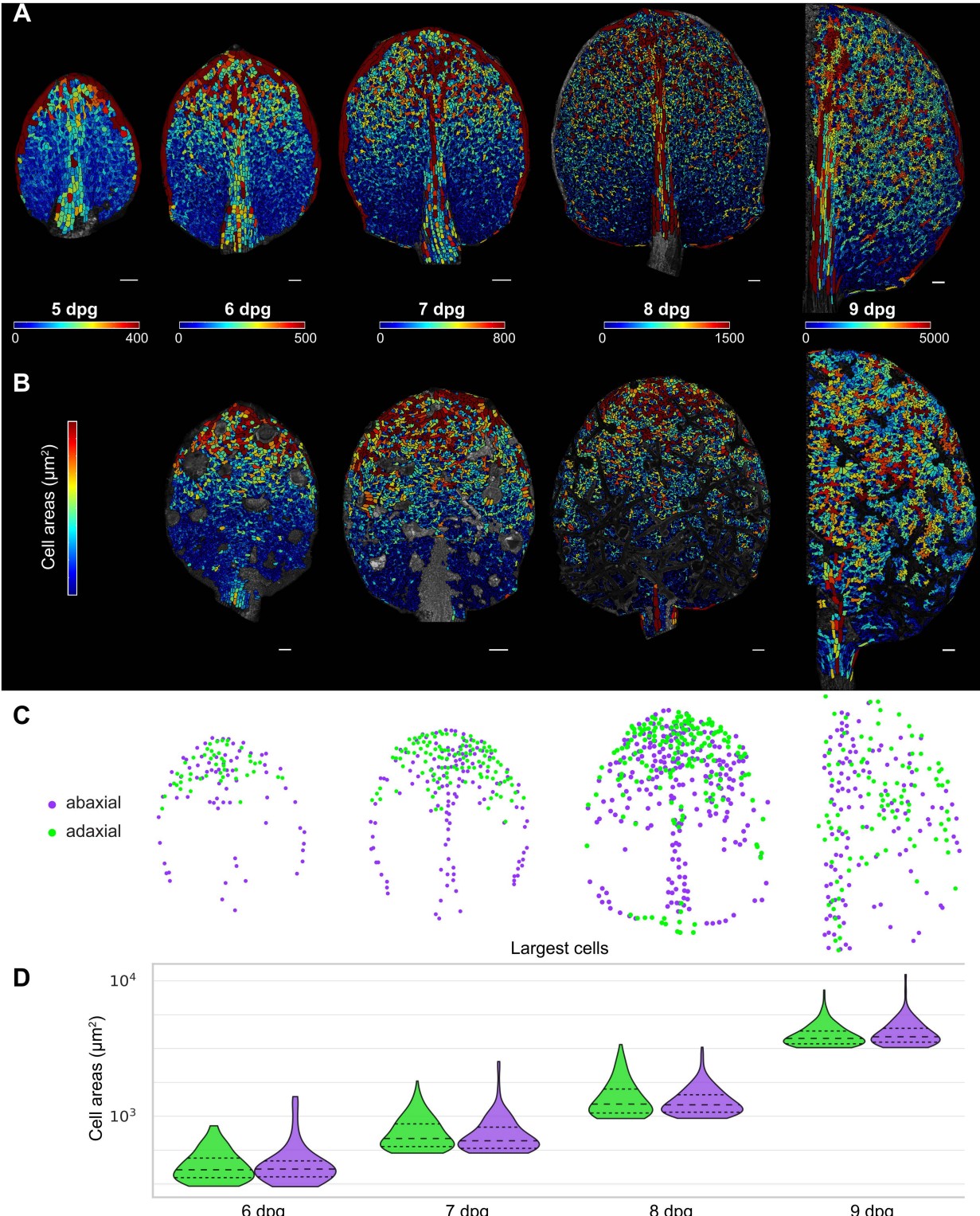

**Fig 3. Cell-size patterning occurs as a basipetal wave simultaneously on the adaxial and abaxial sides of the leaf. (A–B)** Cell area heat maps (in µm²) of leaf 1 or 2 at different stages of development on (A) the abaxial side and on (B) the adaxial side of the same leaf at 5, 6, 7, 8, and 9 dpg (half leaf). Unsegmented regions on adaxial leaves correspond to trichomes, which were not considered in this analysis. Each stage is associated with a

distinct heat map color range. Scale bars represent 50 µm at 5 and 6 dpg, and 100 µm at 7, 8, and 9 dpg. **(C)** Spatial positions of the largest cells (those above an area threshold, see below) on the abaxial (purple points) and adaxial (green points) sides of the same leaf at 6, 7, 8, and 9 dpg. Area thresholds for each leaf were determined from the 98th percentile cell area of the abaxial side. (Note that the midrib and margin cells are included in these overlays of large cell positions.) **(D)** Violin plots of cell areas (in µm$^2$) of the largest cells (as defined in (C) and excluding margin and midrib cells) on abaxial and adaxial sides of leaves at different developmental stages. Abaxial and adaxial sides are from the same leaf. See also S3 Fig for the leaves shown to scale. The underlying data for this figure can be found at Open Science Framework (osf.io), https://doi.org/10.17605/OSF.IO/RFCWS.

To quantify the variations in the number of large cells precisely, we quantitatively defined leaf giant cells on the basis of a cell area threshold. Specifically, we first classified pavement cells and stomata using a Support Vector Machine classifier based on features of cell shape (see Materials and methods, S9 Fig). Next, a cell-size threshold was established in the mature sepal and in the leaf, at both 9 and 25 dpg, using the *atml1-3* mutants, which are known to have very few giant cells in sepals (see Materials and methods, Figs 1C and S9). Those cells in the 9- and 25-dpg leaves as well as in the sepal that exceeded their associated threshold were categorized as giant cells (see cell-type classification outcomes in S10 and S11 Figs). On the basis of this definition, we performed a quantitative comparison and statistically compared the number of giant cells per unit area among genotypes in leaves. Two-sample, two-tailed *t* tests showed that in the 9-dpg leaf and the mature leaf, wild type had significantly more giant cells than *lgo-2* (9 dpg: $p = 0.002$, 25 dpg: $p = 0.003$), *dek1-4* (9 dpg: $p = 0.002$, 25 dpg: $p = 0.002$), *atml1-3* (9 dpg: $p = 0.002$, 25 dpg: $p = 0.005$), and *acr4-2* (9 dpg: $p = 0.010$, 25 dpg: $p = 0.044$). Conversely, *LGO-OX* had significantly more giant cells than wild type (9 dpg: $p = 0.001$, 25 dpg: $p = 0.003$). However, no statistically significant difference in the number of giant cells per unit area was observed between wild type and *ATML1-OX* (9 dpg: $p = 0.213$, 25 dpg: $p = 0.75$). Because the giant cells in *ATML1-OX* are so much bigger than wild-type giant cells, each of the giant cells in a given area of *ATML1-OX* leaf takes up a large amount of space, resulting in few giant cells per unit area despite the fact that most of the unit area is occupied by giant cells. We sought to quantify what was apparent visually by comparing the fractional area occupied by giant cells between *ATML1-OX* and wild type and found that the fractional area occupied by giant cells was significantly higher in *ATML1-OX* (9 dpg: $p < 0.005$, 25 dpg: $p < 0.005$). Thus, in *ATML1-OX* the number of giant cells is not changed, but the fractional area covered by giant cells is increased.

Collectively, the similarities in the variation between the number of giant cells in the leaf and the sepal indicates that the sepal giant cell specification pathway also regulates the formation of giant cells in leaves.

## Giant cell mutants affect the entire cell-size distribution

We observed that not only are giant cells affected in these mutants, but other aspects of the cell-size distribution are also affected. For example, the number of medium-sized cells in *lgo-2* and *dek1-4* is reduced in addition to the number of giant cells (Fig 4A–4D) and, correspondingly, the number of small cells is increased in these mutants. To statistically analyze the difference in cell-size distributions, we conducted a principal coordinate analysis based on the Wasserstein distances between cell-size distributions (termed Wasserstein distance plot), which showed the difference between leaf samples according to their cell-size distributions on a 2-dimensional plane (S4H, S6H, and S8C–S8F Figs, see Materials and methods). Samples clustered according to genotype, indicating that genotype controls the cell-size distribution. We observed a progressive increase in the number of giant cells along the first principal coordinate V1 from *lgo-2* mutants to *ATML1-OX* and *LGO-OX* (S4H and S6H Figs). *ATML1-OX* and *LGO-OX* were distant from each other in this plot, which might partly reflect the fact that *LGO-OX* has more giant cells, whereas *ATML1-OX* has fewer but larger giant cells. When we created the combined Wasserstein distance plot using normalized cell-size distributions from both 9- and 25-dpg leaves (see Materials and methods), the samples continued to group according to genotype rather than developmental stage, further supporting the idea that these genes have affected the cell-size distribution by 9 dpg (Fig 4E). Thus, we conclude that these genes affect the entire cell-size distribution.

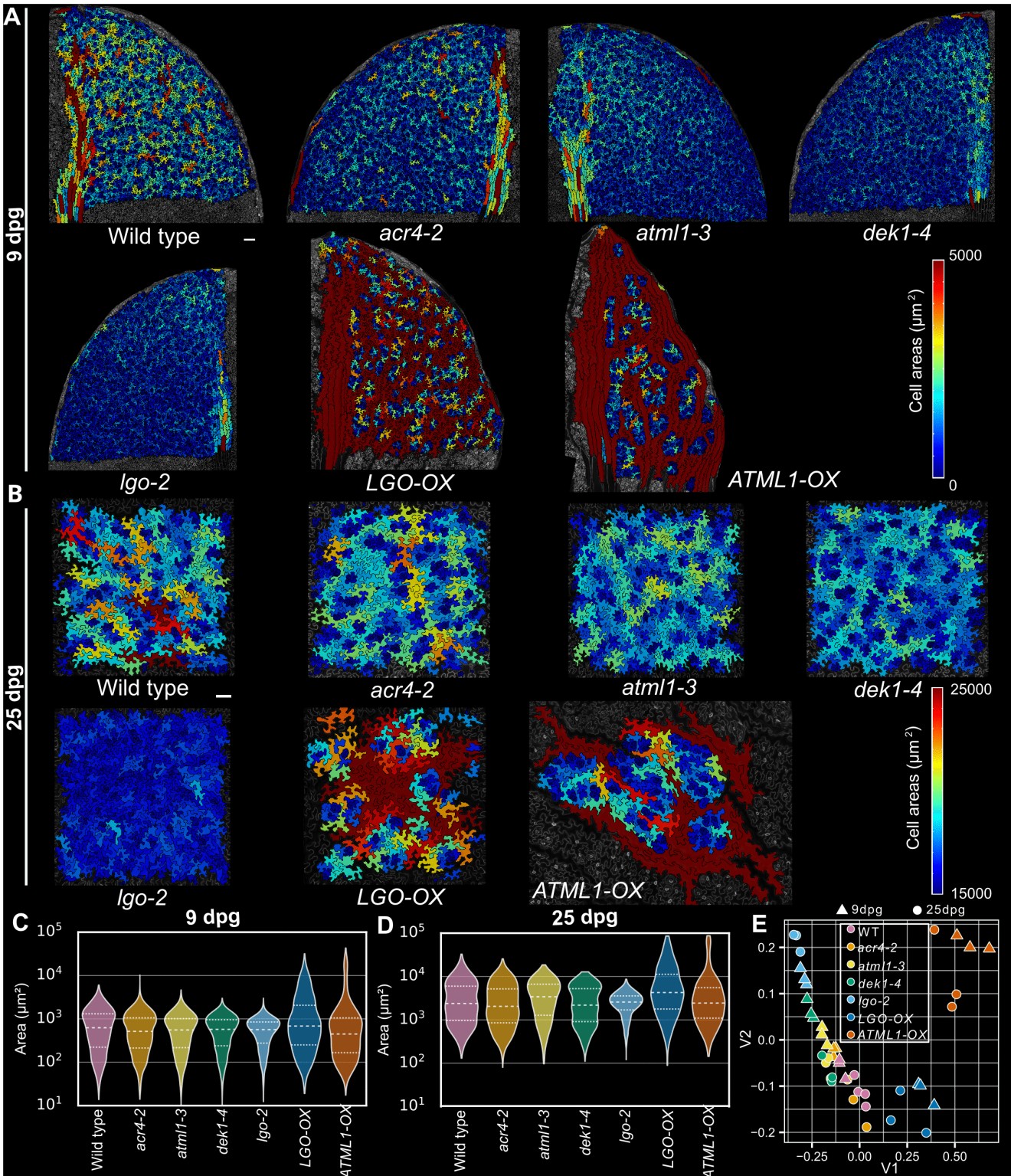

**Fig 4. The sepal giant cell specification pathway also patterns cell size in leaves. (A, B)** Cell area heat maps (in µm²) of the upper abaxial quadrant of leaf 1 or 2 at 9 dpg (A) and of a section on the abaxial side of leaf 1 or 2 at 25 dpg (B) for the following genotypes: wild type, *acr4-2*, *atml1-3*, *dek1-4*, *lgo-2*, LGO-OX (*pATML1::LGO*), and *ATML1-OX* (*pPDF1::FLAG-ATML1*). Scale bars represent 100 µm. Cell area heat maps of other replicates

are shown in S4–S7 Figs. **(C, D)** Violin plots of cell area distributions on a log$_{10}$ scale for 9-dpg replicates (C) and for 25-dpg replicates (D). Stomata were classified and removed in (C, D). Violin plots of individual replicates are shown in S8A and S8B Fig. **(E)** Wasserstein distance plot of normalized cell-size distributions (see Materials and methods) for all 9- and 25-dpg replicates displayed as Euclidean distances embedded in 2D. The 25-dpg replicates are indicated by circular dots and 9-dpg replicates by triangular dots. The Wasserstein distance plot for 9- and 25-dpg and Wasserstein statistical tests among replicates are shown in S4, S6 and S8 Figs. Associated with S4–S8 Figs. The underlying data for this figure can be found at Open Science Framework (osf.io), https://doi.org/10.17605/OSF.IO/RFCWS.

However, some differences in the cell-size distribution are apparent between 9-dpg and mature 25-dpg leaves. Firstly, at 9 dpg, *dek1-4*, and *lgo-2* mutants are very similar; however, in the fully mature 25-dpg leaves, the *lgo-2* cell-size range is notably smaller than that in the *dek1-4* mutant (Fig 4A–4D), suggesting that *lgo-2* cells continue to divide after 9 dpg. In addition, the small cells in *lgo-2* mutants were more uniform in size than all of the other genotypes because the typical small stomatal lineage cells that encircle the stomata in mature leaves were fewer in *lgo-2* (Fig 4A–4D). This altered cell-size distribution relates to the previous finding that *LGO* affects pavement cell differentiation in these stomatal lineage ground cells and that cells undergo division for a longer time in the absence of LGO [19]. Secondly, although at 9 dpg the *LGO-OX* giant cells were slightly smaller than the *ATML1-OX* giant cells, at 25 dpg, the *LGO-OX* giant cells were nearly equivalent in size to *ATML1-OX* giant cells (Fig 4A–4D). In addition, we observed that more pavement cells were larger in *LGO-OX*, whereas only a few cells became giant in *ATML1-OX* (Fig 4A–4D). *ATML1-OX* leaves had a few connected giant cells separating large islands of small cells, whereas *LGO-OX* leaves showed more giant cells interspersed among smaller clusters of small cells (Fig 4A and 4B and S10 and S11) ). These phenotypic differences might reflect either inherent differences in ATML1 and LGO activities or the fact that *ATML1* and *LGO* overexpression transgenes are under the control of different promoters that might have differences in activity at different developmental stages.

## Relationship between the size and shape of cells and organs

In plants, compensation is the process by which organ size is maintained when cell number is altered by an accompanying change in cell size [31]. We observed compensation in the leaf giant cell mutants (S12 Fig). Mature leaves of the mutants *acr4-2*, *atml1-3*, *dek1-4,* and *lgo-2* are similar in size to wild-type leaves. Having fewer giant cells is compensated by having more small pavement cells (Fig 4). However, *ATML1-OX* and *LGO-OX* mature leaves, which have much larger cells (see, e.g., Fig 4B and 4D), are smaller than wild type (S12N–S12P Fig). Therefore, only partial compensation for having fewer cells by having larger cells is observed in *ATML1-OX* and *LGO-OX* plants.

Additionally, *ATML1-OX* leaves are narrower than those of wild type and *LGO-OX* (S12A, S12F, S12G, S12I, S12N, and S12O Fig). We also observed that giant cells are more directionally elongated in *ATML1-OX* than in other genotypes (Figs 4A, 4B, S4F, S4G, S5F, S5G, S6F, S6G, S7G and S7H), reflecting the elongated shape of the leaf. This suggests the existence of a relationship between giant cell shape and leaf morphology. Likewise, wild-type cauline leaves are both narrower and more elongated than wild-type rosette leaves, and also have more anisotropic elongated giant cells than in rosette leaves (S1 Fig). This observation supports the idea that cell shape reflects the anisotropy of the growing tissue [32].

## Spatial patterning of giant cells within the leaf blade

In wild-type plants, giant cells vary in position from sepal to sepal and from leaf to leaf [10–12]. An open question has been whether the spatial organization of giant cells is random, or whether there is an underlying order. Classically, many specialized cell types such as stomata and trichomes are spaced such that they are not in direct contact to one another [23,33]. Giant cells are frequently adjacent to each other and, therefore, it is clear that there is not a strong lateral inhibition between them. We set out to determine firstly whether giant cell position is correlated with underlying vasculature and secondly, how giant cells are spatially positioned relative to one another.

## Giant cells are not preferentially positioned overlying the vasculature

We wondered whether giant cell positioning was correlated with the position of leaf vasculature for two reasons. Firstly, we observed that large, highly endoreduplicated cells overlie the midrib of the leaf, extending all the way to the leaf tip (S13A Fig). We wondered whether giant cells might be similarly preferentially located over the other veins. Secondly, we observed that large, highly endoreduplicated cells often appear to "peel" away from the midrib, as if following vascular branches (S13A Fig). This phenomenon is most common in *ATML1-OX* leaves (S13C–S13F Fig). To investigate whether giant cells overlie veins, we traced the veins from the original confocal image onto the heat map of cell area for a 9-dpg wild-type half leaf and four *ATML1-OX* half leaves. We found that many giant cells do not overlie the vasculature (S13B–S13F Fig). Specifically, we noted that the points where giant cells peel off the midrib often do not align with where veins extend from the midvein. Furthermore, the orientation of giant cells do not follow the direction of the veins (S13B–S13F Fig). Instead, veins in *ATML1-OX* plants frequently pass through patches of small cells (S13C–S13F Fig). We conclude that vascular and giant cell patterns are not obviously correlated.

## Giant cells are clustered more often than expected by chance

A cell-autonomous and stochastic mechanism has been proposed to explain giant cell formation in the sepal [12]. However, it remains unknown whether giant cells are randomly arranged within the tissue. To statistically assess the randomness of the pattern, we needed a random reference (or null model) to compare with our experimental replicates. Previous studies addressing this problem considered cells as points [26,34], or used a regular hexagonal grid to build a null model [35]. In our case, these assumptions are not applicable due to the complexity of giant cell shapes and the heterogeneity of cell shapes and sizes that affect cellular arrangements [36]. Therefore, we used the dmSET image-based method [36,37] to generate randomized tissues from the real segmented images (Figs 5A and S14), allowing to randomly shuffle cell positions by preserving cell sizes and shapes of the original tissues (S15 Fig, see Materials and methods). We generated 400 randomized tissues for each biological replicate for both the wild-type sepal and 25-dpg leaf.

Several measures (S16 Fig), such as the mean number of giant cell neighbors per giant cell, which captures the amount of contacts between giant cells, were computed in the experimental data and the corresponding randomized tissues. To statistically assess the randomness of the giant cell pattern, these measures in the real biological tissues (segmentation) were compared with the same measures in all the randomized tissues (randomizations), which formed a null distribution (Figs 5A and S16).

In the randomized tissues, cell sizes were well preserved, but cell shapes were affected in the leaf (Figs 5A, 5B, S15D, and S15E). To ensure that these shape artifacts did not introduce bias in our analyses, we tested our method on a randomly selected population of cells in both leaves and sepals (see Materials and methods) and confirmed the absence of significant bias in the null models (S17 Fig). Additionally, we reconstructed the original leaf tissues with shape artifacts similar to those in the randomized tissues (S18 Fig), and found that giant cell connectivity was largely preserved and that the results remained consistent (S18 Fig) (see Materials and methods).

For both wild-type 25-dpg leaves and mature sepals, when considering the six pooled replicates, the mean number of giant cell neighbors per giant cell was greater than in a randomized null model, and the null hypothesis could be rejected ($p < 0.05$) (Fig 5C). This result shows the presence of clustering among giant cells both in the leaf and the sepal. It was less probable to find isolated giant cells, and more probable to find giant cells in contact with two or more other giant cells compared with what was expected by chance (Fig 5D). Similar results were found in the leaf using an alternative randomization method we developed based on cutting and merging cells (S19 Fig). The nonrandom pattern of giant cells was also supported by the analysis of other spatial measures (S16 Fig). The similar distribution of the number of giant cell neighbors in leaves and sepals (Fig 5C and 5D) reflects a similar spatial organization, supporting the idea of common patterning mechanisms.

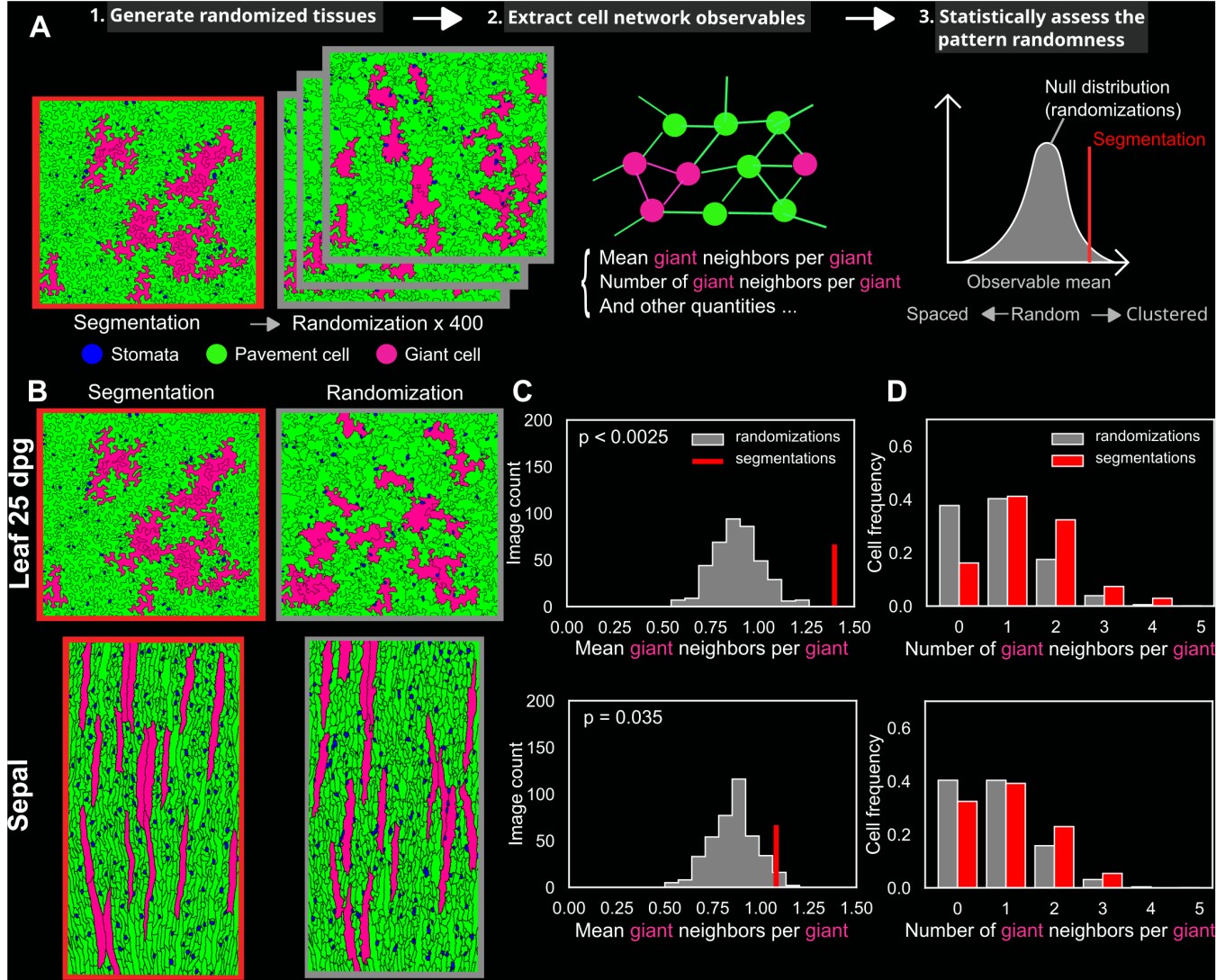

**Fig 5. Giant cells are more clustered than expected in a randomized null model both in the wild-type leaf and sepal. (A)** Scheme summarizing the method used to assess the randomness of the cellular patterns. Each segmentation is computationally randomized using the dmSET method into 400 randomized tissues where cell positions (and orientation in the case of the leaf) have been randomly shuffled (left; see Materials and methods). To statistically assess the extent to which the segmented image shows a random giant cell pattern, a quantitative observable (middle) extracted from the segmentation is compared with the same observable computed in all randomized tissues, forming the estimated 'null distribution' (right). **(B)** Example of a representative segmentation of a wild-type leaf 25 dpg (top left) and a wild-type sepal (bottom left) and one of their randomized tissue (randomization) images (right). **(C)** Mean number of giant cell neighbors (also referred to as giant neighbors) per giant cell (also referred to as giant) in leaves (top) and sepals (bottom). The value extracted from the segmentations (in red) was statistically tested against all the values extracted from the 400 pooled randomizations (in gray). The mean number of giant cell neighbors per giant cell is higher than expected in a randomized null model, and the null hypothesis can be rejected (p-value <0.05), indicating that giant cells are clustered. **(D)** Distributions of the number of giant cell neighbors for all giant cells found in all replicates of segmentations (in red) and randomizations (in gray) in leaves (top) and sepals (bottom). Total number of giant cells counted (excluding giant cells at the image border) in the analysis: $n = 68$ (leaf, segmentations), $n = 68 \times 400$ (leaf, randomizations), $n = 74$ (sepal, segmentations), $n = 74 \times 400$ (sepal, randomizations). See also S14, S15, S16, S21, and S22 Figs. The code and data associated with this figure can be found at Open Science Framework (osf.io), https://doi.org/10.17605/OSF.IO/RFCWS.

## Different cell sizes are organized into different spatial patterns

To investigate whether the clustered pattern is exclusive to giant cells, we applied the same analysis to distinct sub-populations of pavement cells in the leaf tissues. Four populations of pavement cells were defined: giant cells (Fig 6A–6C), middle-sized cells (Fig 6D–6F), small cells (Fig 6G–6I), and a control population of randomly selected pavement cells of any size (Fig 6J–6L). In contrast to the clustered pattern of giant cells (Fig 6A–6C), middle-sized pavement cells exhibited a more random organization (Fig 6D–6F; the null hypothesis could not be rejected, with $p = 0.195$). Conversely, small pavement cells showed a clustered organization (Fig 6G–6I), because the mean number of neighbors between small pavement cells significantly exceeded the value observed in the randomized tissues. Notably, these small cells were clustered around the stomata, and their spatial arrangement is probably a consequence of the stomatal patterning process. A random cellular pattern was found in randomly selected pavement cells, as expected (Fig 6J–6L). Overall, these analyses highlight a relationship between pavement cell size and cell spatial organization within the tissue. Furthermore, these findings underscore the distinctive clustered arrangement of giant cells in comparison to middle-sized and randomly selected pavement cells.

It was previously shown that, except for cells with low neighbour numbers, pavement cells mostly follow the theoretical topological laws expected from space-filling (i.e. entropic) considerations, with larger cells being on average surrounded by smaller ones in young *spch* leaf tissues [38]. Our analyses on more mature wild-type leaves reveal that larger cells are surrounded by larger cells (and have fewer neighbors) than what is expected by purely random space-filling given by our null model (Figs 6 and S18D).

## A cell-autonomous stochastic model can recapitulate giant cell clustering

To investigate how giant cell clustering emerges during leaf and sepal epidermal development, we wondered whether the existing cell-autonomous and stochastic model for giant cell specification in sepals [12] could also recapitulate the clustered feature of the giant cell pattern. In this multicellular computational model, the concentration of ATML1 stochastically fluctuates and is regulated by a self-catalytic feedback loop. ATML1 regulates the expression of a downstream cell-cycle regulator target (Fig 7A). At the end of a cell cycle, a cell either divides or endoreduplicates if the ATML1 target exceeds a specific threshold during the G2 phase. We used this model [12] to investigate the resulting spatial organization of giant cells in simulated tissues (Fig 7A and 7B; see Materials and methods).

To assess the randomness of the simulated giant cell pattern, we applied the same method as in the experimental images (Fig 5A) to images of the final simulation time point (Fig 7B and 7C). Giant cells were also defined by a size threshold, which was established such that all cells of ploidy 16C or above were considered to be giant (see Materials and methods). We observed that the mean number of giant cell neighbors per giant cell was greater than expected if giant cells were randomly distributed ($p < 0.05$, Fig 7D), showing that the current cell-autonomous model can also produce a clustered giant cell pattern. Furthermore, the distribution of the number of giant cell neighbors per giant cell (Fig 7E) was similar to the distribution observed in the experimental sepals (Fig 5D, bottom). This raises the question of what mechanisms are responsible for cell clustering in a cell-autonomous, multicellular model of dividing cells.

## Cell division contributes to the clustering of giant cells

To understand how the giant cell clustering behavior emerges in our computational model, we analyzed how the cellular spatial pattern changes over time within the tissue. We hypothesized that the initial giant cell pattern arises randomly throughout the epidermis, due to the stochastic nature of ATML1 concentration fluctuations that trigger endoreduplication, and occasionally lead to giant cell contacts. As non-giant cells continue to divide, giant cells would appear more clustered in the fully grown tissue. To test this hypothesis in our simulations, we selected the first-arising giant cells (see Materials and methods) and quantified their spatial organization both at an early time point and at the end of the simulation (Figs 7F,S20A, and

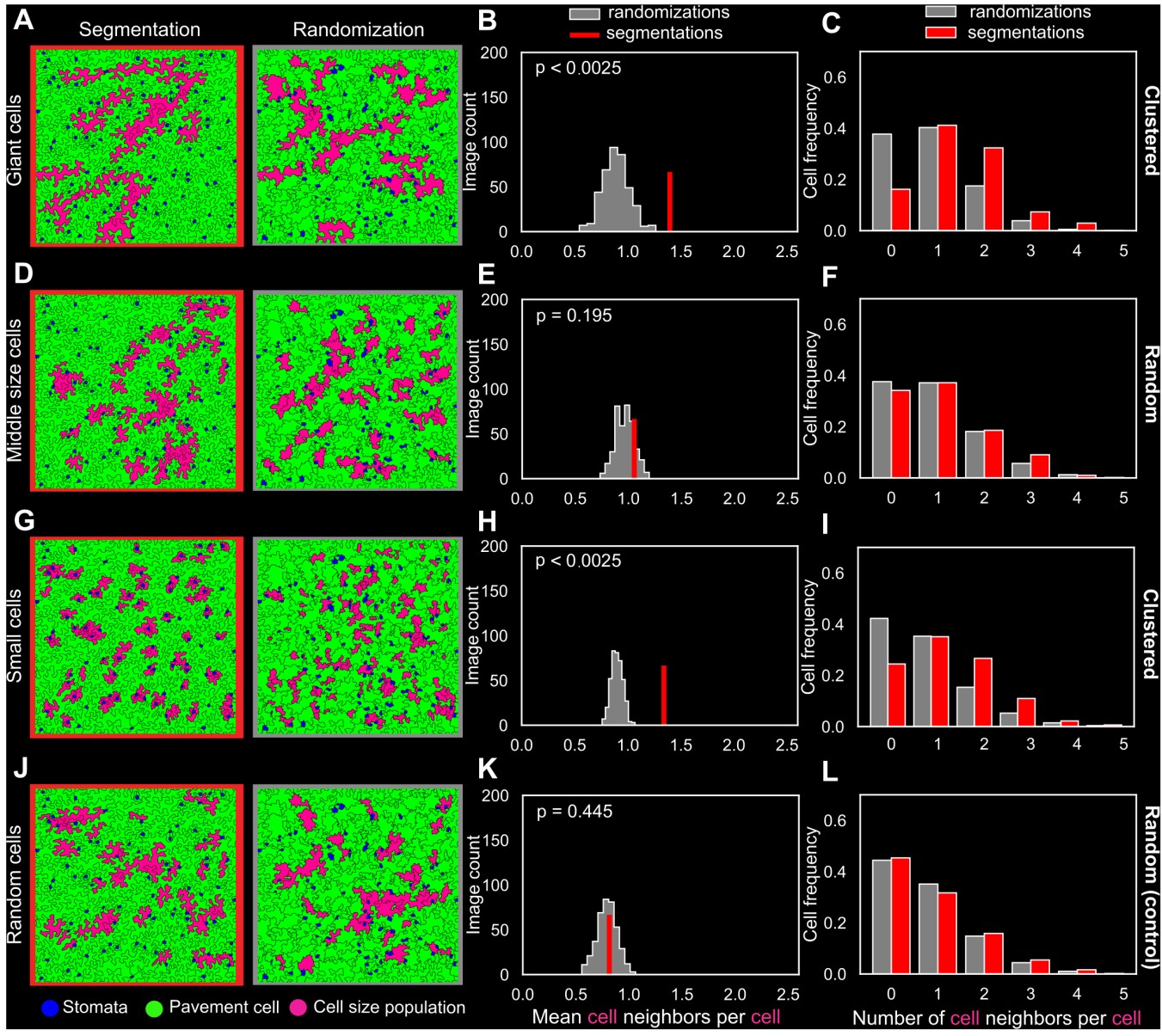

**Fig 6. Different cell sizes display different spatial patterns in the wild-type leaf.** The method used to assess the randomness of the giant cell patterns (Fig 5) was applied here on different pavement cell-size populations within the mature 25-dpg leaf: **(A–C)** giant, **(D–F)** mid-size (around 5,000 µm²), **(G–I)** small (smallest pavement cells), and **(J–L)** random (randomly selected pavement cells). The number of cells in each category was determined such that the total cell area of the cell population was approximately equal to the area occupied by the giant cells. (A, D, G, J) Example of representative segmentation of a 25-dpg wild-type leaf (left) and one of its corresponding randomized tissues (right), where cell locations have been computationally shuffled. Cells colored in magenta represent the cells within the studied pavement cell-size population. (B, E, H, K) Mean number of cell neighbors per cell within the same-size population. (B) The mean number of giant cell neighbors per giant cell is higher than expected by chance ($p < 0.05$), indicating that giant cells are clustered. Same data as in Fig 5C, top. (E) Middle-size cells are less clustered than giant cells and more randomly organized (the null hypothesis cannot be rejected, $p = 0.195$). (H) The mean number of small cell neighbors per small cell is significantly higher than in the randomized tissues ($p < 0.05$), highlighting that small cells form clusters. (K) As expected, the randomly selected pavement cells (with area > 2,000 µm²) show a value that falls right in the center of the null distribution ($p = 0.445$). (C, F, I, L) Distributions of the number of cell neighbors belonging to the studied cell population per cell of that population in the segmentations (in red) and the randomizations (in gray). All six replicates were pooled together. Total number of cells in cell populations counted in the analysis: $n = 68$ (giant cells, segmentations), $n = 68 \times 400$ (giant cells, randomizations), $n = 199$ (middle-size cells, segmentations), $n = 199 \times 400$ (middle-size cells, randomizations), $n = 639$ (small cells, segmentations), $n = 639 \times 400$ (small cells, randomizations), $n = 162$ (random cells, segmentations), $n = 162 \times 400$ (random cells, randomizations). The code and data associated with this figure can be found at Open Science Framework (osf.io), https://doi.org/10.17605/OSF.IO/RFCWS.

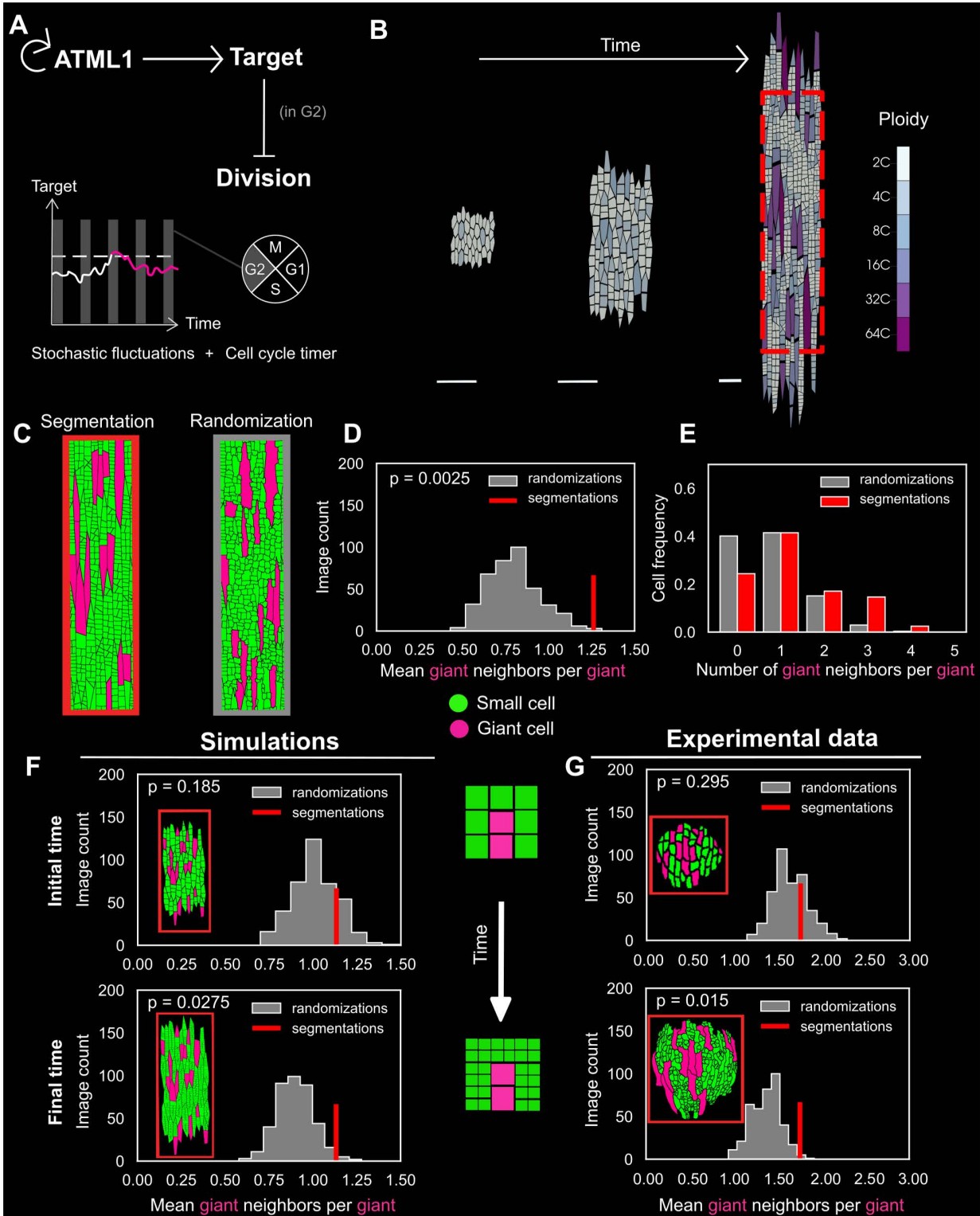

**Fig 7. A cell-autonomous stochastic model can recapitulate giant cell clustering because of cell divisions of surrounding cells. (A)** Cartoon of the computational model for giant cell patterning. ATML1 activates a target (LGO). If the target is above a certain threshold during the G2 cell-cycle phase (line changing from white to magenta in the cartoon of the time course), it prevents cell division and instead drives entry of the cell into

endoreduplication and giant cell formation. **(B)** Snapshots of the simulated growing sepal, at three different time points. Color codes indicate the cell ploidy levels. Scale bars represent the same size in arbitrary units. **(C)** A rectangular section of the simulation output (e.g., see rectangle shown in (B)) is used to quantify the giant cell pattern. "Segmentation" refers to one simulation output (left) and "Randomization" to one randomization of the simulated output (right). Giant cells, labeled in magenta, were defined by a size threshold (see Materials and methods). **(D)** Mean number of giant cell neighbors (also referred to as giant neighbors) per giant cell (also referred to as giant) in the simulations (called segmentation in red) and in their randomizations (in gray). The mean number of giant cell neighbors per giant cell is higher than expected in a randomized null model ($p < 0.05$), indicating a clustered pattern of giant cells. **(E)** Distribution of the number of giant cell neighbors per giant cell. The results obtained in (D, E) are comparable with the results in experimental sepal replicates (Fig 5C and 5D). Five simulation outputs with five different initial conditions were performed and combined for the analysis. Total number of giant cells (excluding giant cells at the image border) counted in the analysis: $n = 42$ (segmentations), $n = 42 \times 400$ (randomizations). **(F, G)** Statistical assessment of the randomness of the giant cell pattern (comparing the "segmentations" in red with the randomized tissues in gray) at initial time (top) and final time point (bottom) in (F) the simulations (at $t = 55$ and $t = 135$) and (G) the real tissues (at stage 4 and stage 9). At the initial time point, the null hypothesis could not be rejected but the mean giant cell neighbors per giant cell became significantly greater than in a randomized null model ($p < 0.05$) at the final time point. To the right of panel (F), a cartoon represents two neighboring giant cells surrounded by an increasing number of cells as the tissue develops. All five replicates (in simulations) and three replicates (in experimental data) were pooled. Total number of giant cells counted in the analysis: $n = 83$ (simulations, segmentations), $n = 83 \times 400$ (simulations, randomizations), $n = 49$ (experimental data, segmentations), $n = 49 \times 400$ (experimental data, randomizations). The dataset in (G) was also used for an independent analysis in Hervieux and colleagues 2016. See randomization snapshots related to this figure in S20 Fig. The code and data associated with this figure can be found at Open Science Framework (osf.io), https://doi.org/10.17605/OSF.IO/RFCWS.

S20B). We found that these giant cells were more randomly distributed at the initial time point, as the null hypothesis could not be rejected ($p = 0.185$, Fig 7F), whereas they were more clustered compared with the randomized tissues at the final time point ($p < 0.05$, Fig 7F). Indeed, although the giant cell contacts were preserved over time (red bar in Fig 7F), we observed a shift in the null distribution of the mean number of giant neighbors per giant cell between the initial and the final time point (Fig 7F). As new cells arise from cell division, the number of potential cellular configurations (i.e., the number of possible spatial cellular arrangements) increases, which decreases the probability of observing giant cell clusters under a random model where all cells have random positions. Therefore, even if giant cell contacts are preserved, their arrangement in the context of the entire tissue becomes more clustered over time.

To investigate the emergence of the giant cell spatial pattern over time in real tissues, we used time-lapse data of developing sepals [39], where cells were tracked over time, and we similarly quantified the patterns of the first-arising giant cells at the first available time point (sepal at stage 4, 24-h time point) and a later one (sepal at stage 9, 120-h time point; see Materials and methods) (Figs 7G, S20C, and S20D). Similar to the simulations, we observed that giant cells were more randomly distributed in younger sepals and were more clustered in the more developed sepals when compared with the randomized tissues (Fig 7G). This analysis indicates that the stochastic and cell-autonomous model is a plausible model to explain the spatial organization of giant cells. Moreover, it shows that cell clustering can emerge in a growing tissue without the need for cell–cell communication but instead as a result of cell divisions.

## Discussion

We investigated pavement cell-size patterning in the Arabidopsis leaf epidermis. We found that the same genetic pathway that controls giant cell formation in sepals also controls cell size and giant cell formation in the leaf. Specifically, the receptor-like kinase ACR4, the transcription factor ATML1, the calpain protease DEK1, and the CDK inhibitor LGO are important for the formation of leaf giant cells. Just as in the sepal, overexpression of *LGO* results in an increased number of giant cells and overexpression of *ATML1* leads to a larger area occupied by giant cells. Although giant cells are only present on the abaxial epidermis of sepals, they are present on both the abaxial and adaxial surfaces of leaves. We observed that giant cells are scattered across the surface, sometimes in contact with one another, in both leaves and sepals. Our analysis demonstrated that giant cells are more likely to be in contact than in a randomized tissue null model in both organs.

Many patterning systems rely on cell–cell communication to generate proper spacing [7,40], and the emergence of clustered patterns in certain cell types is often attributed to cell–cell communication mechanisms in static tissues [36,41,42]. However, giant cell specification occurs within the context of tissue growth and cell division. Therefore, it is

important to consider the influence of these dynamic factors as well. We revisited our previous cell-autonomous model for giant cell specification in which ATML1 stochastically fluctuates, and confirmed that giant cell clustering could arise in that model as a result of cell division, without the need for cell–cell communication. To understand how clustering emerges, we tracked giant cells from their initial emergence both in our modeled tissue and in published experimental time-lapse data of growing sepals [39], and analyzed the evolution of their spatial pattern over time. This analysis suggested that the giant cell pattern initially arises randomly in space in the primordium and becomes more clustered in the context of the fully grown tissue. Therefore, the decrease in the randomness of the giant cell pattern over time appears to be caused by the division of cells surrounding the giant cells, including dividing stomata lineage cells. Rather than resulting from active cell–cell communication, giant cell clustering reflects the history of cell division and tissue growth. Nevertheless, in a proliferating tissue, other mechanisms might operate at the same time that result in giant cell clustering. For instance, correlative effects on cells belonging to the same lineage might influence cell fate decisions, e.g., due to the inheritance of molecular factors from mother to daughter cells [43]. Although we show that giant cell clustering can occur as a result of surrounding cell divisions, whether or not cell–cell communication or other patterning mechanisms may also exert some effect on giant cell clustering is yet to be determined and warrants further study.

In the past, researchers have attempted to increase organ size by increasing cell size by promoting endoreduplication, but these efforts have not been successful [44]. This is because compensation occurs, in which smaller cell size is accompanied by an increase in cell number, so that organ size is relatively conserved [45,46]. Consistent with this, we observed that leaf 1 or 2 of wild type, *atml1-3*, and *lgo-2* plants are approximately the same size at maturity. Furthermore, instead of having larger leaves, the *ATML1-OX* and *LGO-OX* genotypes that have larger cells actually have slightly smaller leaves than the wild type at maturity. These observations are consistent with what is observed for sepals where *ATML1-OX* and *LGO-OX* sepals are slightly smaller than wild-type sepals [10,45]. We have previously shown that mitotic division substitutes for endoreduplication to compensate and maintain organ size in mutants lacking giant cells [10]. Our images suggest that this mechanism also operates in leaves.

Although giant cell number does not greatly influence organ size, organ shape is altered in sepals and leaves. *ATML1-OX* and *LGO-OX* sepals are narrower than those of wild type and curve outward, so that the bud opens prematurely [11]. We speculate that the anisotropy of sepal giant cells drives the change in sepal shape. In *ATML1-OX* leaves, where giant cells are highly anisotropic, we observe a similar change in leaf shape, in which *ATML1-OX* leaf 1 and 2 are more pointed and oblong compared with the rounded wild-type leaf 1 and 2. By contrast, giant cells in *LGO-OX* leaves are more isotropic and similar to wild-type giant cells; correspondingly *LGO-OX* leaves are more rounded. Our results suggest that ATML1 is sufficient to induce anisotropic cell growth, whereas LGO is not. Tang and colleagues (2023) have shown that the change in shape between rounded juvenile rosette leaves and more elongated adult rosette leaves is accompanied by the appearance of highly anisotropic giant cells at the leaf base [32]. However, they showed that loss of these directional, elongated giant cells does not change adult rosette leaf shape; the adult rosette leaf 7 remains elongated in *lgo-2* leaves where giant cells are not present [32]. Thus, the relationship between giant cell shape, anisotropic growth, and organ shape is complex. Further work at the single-cell level will be needed to elucidate the influence of giant cells on the shapes of different tissues.

Despite the similarities between cell-size patterning in leaves and sepals, subtle differences also exist. Firstly, the distribution of epidermal cell sizes in the leaf is broader than in the sepal, where cells are fairly uniformly small except for a scattering of giant cells (Fig 2E). Secondly, leaves have giant cells on both abaxial and adaxial blades, whereas sepals have giant cells only on the abaxial side and not on the adaxial side that faces the petals. The petal blade does not have giant cells on either abaxial or adaxial sides [47]; thus, sepals might be an organ whose identity is transitional between vegetative and floral organs [48]. We observe a similar phase change in the anisotropy of giant cells. Rosette leaf giant cells are jigsaw puzzle-piece shaped and relatively isotropic. Later in the plant life cycle, giant cells in cauline leaves begin to be more anisotropic along the proximal–distal axis and start to resemble sepal giant cells. This supports the hypothesis

that cauline leaves represent an intermediate state between rosette leaves and sepals [49]. Finally, sepal giant cells are highly anisotropic along the proximal–distal axis. Although sepals and leaves have notable yet subtle differences in cell size, cell-size patterning is regulated by the same developmental pathway in both organs.

The genetic pathway that regulates giant cell specification has been co-opted from the epidermal specification pathway, which is a developmental pathway necessary for epidermal and thus plant development [14,50,51]. Without proper epidermal specification, the plant embryo will not progress past the globular stage of development [50–52]. The fact that this fundamental epidermal developmental pathway also patterns giant cells illustrates a common theme in development, namely, that regulatory proteins are commonly reused for more than one developmental process [53].

Taken together, our analysis and theoretical work on patterning during tissue growth highlights that unexpected effects can occur, and that these are difficult to infer from the canonical view of pattern formation arising in a static tissue. In this instance, an initially random pattern of giant cells becomes nonrandom as the surrounding cells divide. Thus, the effects of cell proliferation might also be important to determine the spatial distribution of specialized cell types in other tissues.

## Materials and methods

### Plant growth conditions

All seeds were sown on LM111 soil in pots and were stratified in the dark for 3 days at 4 °C. The pots were then transferred to Percival plant growth chambers set to 60% humidity, 22 °C temperature, and 24-h light provided by Philips 800 Series 32-Watt fluorescent bulbs (f32t8/tl841) (~100 μmol m$^{-2}$ s$^{-1}$). dpg were counted from the time the pots were transferred to plant growth chambers.

### Cloning fluorescent nuclear markers

To create a teal fluorescent (TFP) nuclear marker ubiquitously expressed under the *UBIQUITIN 10* promoter (*pUBQ10::H2B-TFP; pAR393*), an H2B-TFP fusion with an AAAPAAAAAA linker was generated by PCR. TFP was amplified by PCR with primers oAR440 (5′-gct gcc gct cca gct gca gct gcc gct ATG GTT TCT AAG GGA GAA GAA ACT ACT ATG-3′) and oAR438 (5′-cct cga gtc aCT TAT AAA GTT CAT CCA TAC CAT CAG TAG-3′). The lower-case letters in the primer sequences represent linkers, restriction sites, and cloning sequences that were added to the gene sequences. H2B was PCR amplified with oAR369 (5′-CAC CGG ATC CAC AAT GGC GAA GGC AGA TAA G-3′) and oAR439 (5′-agc ggc agc tgc agc tgg agc ggc agc AGA ACT CGT AAA CTT CGT AAC CGC CTT AG-3′). Sequences encoding H2B-TFP were fused via overlapping PCR with oAR369 and oAR438 primers. The H2B-TFP PCR product was cloned into pENTR D TOPO to create the pAR198 entry clone. H2B-TFP was recombined into pUB-Dest with LR Clonase II according to the manufacturer's instructions to generate *pUBQ10::H2B-TFP* (pAR393). pAR393 was transformed into Arabidopsis Col-0 plants expressing the pLH13 *p35S::mCitrine-RCI2A* yellow fluorescent plasma membrane marker [45] via *Agrobacterium tumefaciens* (strain GV3101)-mediated floral dipping [54] and selection with glufosinate-ammonium ("Basta" Neta Scientific OAK-044851-25g).

### Generation of mutant plant lines containing fluorescent plasma membrane and nuclear markers for imaging

Mutant alleles and overexpression transgenes were crossed to plants expressing fluorescent cell-membrane and nuclear markers to obtain plants for imaging. The following mutant alleles were used: *acr4-2*, *atml1-3*, *dek1-4*, and *lgo-2*. In addition, two lines overexpressing either ATML1 (*pPDF1::FLAG-ATML1*) or *LGO* (*pATML1::LGO*) in a Col-0 background were used. All of these alleles/transgenes are in the Columbia-0 (Col-0) accession. *acr4-2* (SAIL_240_B04) contains a T-DNA insertion in the codon of the second of seven 39-amino-acid repeats of the beta propeller extracellular domain, which is upstream of the transmembrane domain and the kinase domain and is therefore presumed to be loss-of-function allele. The *acr4-2* mutant was obtained from Gwyneth Ingram [14], who obtained it from Syngenta [55]. The *atml1-3* allele is a T-DNA insertion in the homeodomain and is a loss-of-function mutant [11]. The *atml1-3* was obtained from the Arabidopsis Biological Resource Center (ABRC; accession number SALK_033408) [11]. The *dek1-4* allele contains a point mutation that changes

a conserved arginine to a cysteine within calpain domain III (ABRC accession CS68904) [11]. Complete loss of function of DEK1 is lethal [50]; therefore, *dek1-4* must retain some function. The *dek1-4* phenotype is recessive and therefore, *dek1-4* is likely hypomorphic [11]. The *dek1-4* mutant was originally isolated in the Landsberg erecta accession and was subsequently back-crossed twice into Col-0 [12]. *lgo-2* contains a T-DNA insertion within the coding sequence of the gene and is a loss-of-function allele [10]. *lgo-2* was obtained from the ABRC (accession number SALK_033905 and is available as a homozygous mutant as accession CS69160). *pPDF1::FLAG-ATML1* (*ATML1-OX*) was obtained from Gwyneth Ingram [56]. *pATML1::LGO* (*LGO-OX*) has been deposited for distribution at ABRC under accession CS69162 [11,45]. *acr4-2*, *atml1-3*, *dek1-4*, *lgo-2*, *pPDF1::FLAG-ATML1*, and *pATML1::LGO* were each crossed to plants expressing both a *p35S::mCitrine-RCI2A* fluorescent plasma membrane marker (pLH13) and a *pUBQ::H2B-TFP* fluorescent nuclear marker (pAR393). The F2 progeny were genotyped for *acr4-2*, *atml1-3*, *dek1-4*, and *lgo-2* (primer sequences in S1 Table) and lines were isolated that were homozygous for these alleles and that also expressed both *pUBQ::H2B-TFP* and *p35S::mCitrine-RCI2A*. We could not obtain *atml1-3* homozygous plants that also contained the *p35S::mCitrine-RCI2A* transgene after crossing, which was probably because the *ATML1* gene was linked to the insertion site of the *p35S::mCitrine-RCI2A* transgene. To obtain plants expressing *p35S::mCitrine-RCI2A* in a homozygous *atml1-3* background, the plasmid containing *p35S::mCitrine-RCI2A* was transformed into *atml1-3* homozygous plants with *pUBQ::H2B-TFP* through *Agrobacterium tumefaciens* (strain GV3101)-mediated floral dipping [53]. A T1 line was chosen that strongly expressed the mCitrine membrane signal and this line was used for future experiments. For the overexpression transgenes *pPDF1::FLAG-ATML1* and *pATML1::LGO*, seeds were collected from F2 plants and F3 plants were genotyped for *pPDF1::FLAG-ATML1* or *pATML1::LGO* (S1 Table). Those F2 plants that produced only F3 plants having *pPDF1::FLAG-ATML1* or *pATML1::LGO* were isolated as homozygous for *pPDF1::FLAG-ATML1* or *pATML1::LGO*, respectively. Those lines with *pPDF1::FLAG-ATML1* or *pATML1::LGO* homozygous and that expressed both *pUBQ::H2B-TFP* and *p35S::mCitrine-RCI2A* were used for imaging.

## Sample preparation for imaging

Leaves and sepals were mounted in 0.01% (v/v) Triton X-100 for imaging. Leaves were imaged between two coverslips, and sepals were imaged on a slide with a coverslip. Curvy leaves were cut with a razor blade to ensure they could be placed flat under the coverslip. Samples were imaged immediately after preparation. Sepals were imaged at stage 14 [57].

## Imaging with confocal microscopy

A ZEISS LSM 710 Axio Examiner confocal microscope with a W Plan-Apochromat 20×/1.0 DIC D-0.17 M27 75 mm water-immersion objective lens was used to image leaf 1 or 2 of the Arabidopsis rosette and mature (stage 14) sepals. A 458 nm laser was used to excite TFP in *pUBQ::H2B-TFP* (collection range 463–500 nm) and a 514 nm laser was used to excite mCitrine in *p35S::mCitrine-RCI2A* (collection range 525–645 nm). Images were captured with a 1× zoom. The gain and laser power varied slightly between images to accommodate slight differences in signal intensity between samples. Each image was composed of several tiles. The dimensions of each voxel were 0.415 μm ($x$) by 0.415 μm ($y$) by 1 μm ($z$).

For the images to compare cell size with nuclear fluorescence on the abaxial and adaxial faces of the same organ (two leaf replicates and three sepal replicates; Figs 2 and S2), the 458-nm laser power and gain used for imaging TFP in *pUBQ::H2B-TFP* were adjusted so that the TFP signal was below saturation and was then held constant for all images.

Leaf areas were calculated from confocal images of entire leaves taken using a 2.5× objective for each 9-dpg and 25-dpg leaf replicate.

## Image processing

Tiles were stitched in the horizontal direction by ZEISS stitching software (overlap of 5% and threshold of 0.7) and in the vertical direction with MorphoGraphX [58,59] using the process "Stacks/Multistack/Merge Stacks" (parameters: method = max; interpolation = linear). Assembled images were saved as MorphoGraphX stack files. A surface mesh was

created in MorphoGraphX from each image to perform segmentation and analysis on the epidermis. First, extraneous parts of the image were removed with the Voxel Edit tool. (Such extraneous parts of the image include trichomes on the adaxial images and pollen grains/nematode eggs on some leaf images.) Then, an image was subjected to Gaussian Blur using the process "Stack/Filter/Gaussian Blur Stack" (parameters: $x=2$; $y=2$; $z=2$). Next, the tissue surface was identified with the process "Stack/Morphology/Edge Detect" (parameters: threshold varied between 2,300 and 7,000 according to individual image brightness; multiplier $=2.0$; adapt factor $=0.3$; fill value $=30,000$). These steps extracted a surface of the leaf. The process "Stack/Morphology/Fill Holes" was applied to some images when holes were apparent in the surface (parameters: $x$-radius $=20$; $y$-radius $=20$; threshold $=10,000$; depth $=0$; fill value $=30,000$). This surface was then used to generate a mesh with the process "Mesh/Creation/Marching Cube Surface" (parameters: cube size $=5$ µm; threshold $=20,000$). The mesh was smoothed with "Mesh/Structure/Smooth mesh" (parameters: number of passes varied between 20–45; Walls Only $=$ no). The mesh obtained was then subdivided with the process "Mesh/Structure/Subdivide" either once or twice depending on its size. In order to obtain the cell membrane signal on the surface, the process "Meshes/Signal/Project Signal" (parameters: min/max distances ranged between 5 and 15 µm; MinSig $=0.0$; MaxSig $=60,000$) was used to project the mCitrine signal from the *p35S::mCitrine-RCI2A* plasma membrane marker onto the mesh at an optimal depth. The depth range yielding the clearest cell membrane signals with minimal distortion was selected. To perform cell segmentation, each individual cell in the leaf was first manually identified with a cell label marking (seed). Using these seeds, watershed segmentation was performed using the process "Meshes/Segmentation/Watershed Segmentation." Adjacent pairs of stomatal guard cells were segmented together to form a single cell; these were classified as stomata. Errors in segmentation were identified and corrected by removing the label for those cells and reseeding. The Heat Map processes computed the cell area and other morphological cell features as well as the position of every cell. The cell area data were exported into a data table file for each image. In addition, other cellular shape features were computed and exported into a data table for the cell type classification ([Materials and methods]: cell type classification).

To analyze the nuclear signal from images of leaves expressing *pUBQ::H2B-TFP*, vertical stitching of tiles (already horizontally stitched with Zeiss stitching software) was performed in MorphoGraphX (max method and linear interpolation). Because the *TFP* reporter was expressed under the *UBIQUITIN 10* promoter, TFP was localized in nuclei of the mesophyll cells in addition to cells of the epidermis. Mesophyll nuclei were removed with the Voxel Edit tool. Nuclei were identified as being from the mesophyll by lining up the nuclear signal images with their corresponding membrane signal images and comparing the nuclei within the bounds of each epidermal cell membrane. When compared with an epidermal cell nucleus, mesophyll cell nuclei were often dimmer and lower down and therefore, excess nuclei were removed according to these criteria so that each epidermal cell had one nucleus. When it was ambiguous which of two nuclei in a single cell was from the mesophyll or epidermis, both nuclei were removed and excluded from the analysis. Segmentation of the nuclei was performed in MorphoGraphX so that the total signal could be calculated for each nucleus. To do so, the confocal image was first subjected to "Stack/Filters/Brighten Darken" (parameter: 1). Next, a gaussian blur was performed using "Stack/Filters/Gaussian Blur" (parameters: $x=1$, $y=1$, $z=1$), followed by a binarization with the process "Stack/Filters/Binarize" (parameters: threshold $=2,000$), which functioned to select pixels above a threshold value to identify the edges of each nucleus. A lower threshold value was chosen so that we could identify the entire nuclei even for dim nuclei. The Voxel edit tool was used to separate nuclei that inflated into one another. We then created a mesh from the binarized image using the process "Mesh/Creation/Marching Cubes 3D" (parameters: cube size $=1$, min voxel $=0$, smooth passes $=3$, label $=0$). To ensure that the mesh covered all fluorescence of each nucleus, we expanded the mesh using "Mesh/Structure/Shrink Mesh" with a negative value (parameter: distance $=-1$). Individual nuclei were manually seeded and then the watershed segmentation was performed with the process "Mesh/Segmentation/Watershed Segmentation" to identify each nucleus. The Heat Map function calculated the total H2B-TFP fluorescence within each nucleus, as a representation of DNA content. To study correlations between total nuclear H2B-TFP signal and cell size, individual cells from

cell area meshes were matched with their constituent nuclei from nuclear signal meshes using MorphoGraphX parent tracking. For the leaf replicates, total nuclear H2B-TFP signal was calculated for as many cells as possible from both the abaxial and adaxial sides. For the sepal replicates, total nuclear H2B-TFP signal was calculated only on the abaxial side and only for the largest cells.

To create the heat maps overlaid with vasculature in S13 Fig, confocal images of leaves expressing *p35S::mCitrine-RCI2A* were used to create surfaces and were segmented as described above to create cell area heat maps. The mCitrine-RCI2A confocal images were found to have signal in the vasculature, so that the trajectories of veins could be traced in images from the abaxial surface of the image. The mCitrine-RCI2A confocal images were transformed around the *z*-axis in MorphoGraphX. For each leaf, the cell area heat map and the mCitrine-RCI2A confocal image transformed around the *z*-axis were aligned in MorphoGraphX and PNG screen captures were taken of each. These PNGs were then loaded into Adobe Illustrator and the veins were traced in white onto the heat maps.

Please note that wild-type 25 dpg leaf replicates 1, 3, and 4, *lgo-2* 25 dpg leaf replicates 1 and 2, and *LGO-OX* replicates 1, 2, and 3 were used for an independent analysis of cell shape (specifically lobeyness) [60].

## Statistical analysis

To analyze the relationship between total nuclear H2B-TFP signal (DNA content) and cell area for the leaves in Figs 2 and S2, linear regressions were performed on R statistical software (https://www.r-project.org/). To compare the total nuclear H2B-TFP signal (DNA content) of the cells of largest area between sepals and leaves, the cell area at the 98th percentile was calculated for each of the three abaxial sepal replicates and these three cell areas were averaged for an area threshold of 4,308 $\mu m^2$. Cell area versus total nuclear H2B-TFP signal (DNA content) was plotted for cells above this 4,308 $\mu m^2$ area threshold for the abaxial sepals and the abaxial and adaxial leaves.

To compare positions of the largest cells on the abaxial and adaxial sides of each leaf at different stages of development (Fig 3), the abaxial and adaxial images were aligned in MorphoGraphX. Then, cell area heat maps were created and the x and y coordinates of the center of each cell were calculated (Fig 3C). Cell area thresholds for each leaf were determined from the 98th percentile cell area of the abaxial side, and the positions of cells above these area thresholds were plotted for the abaxial and adaxial images of each leaf.

R statistical software was used to analyze the cell-size distributions and create the violin plots and Wasserstein plots. The threshold for significance was set to alpha = 0.05. To create the Wasserstein plots, a Wasserstein test was performed between each pair of replicate distributions. A test statistic (also known as Wasserstein distance) and *p*-values were returned for every test. The *p*-values are listed in S8C and S8D Fig. Classical multidimensional scaling was performed to create a 2D coordinate for each replicate distribution based on the Wasserstein distances, and points from these coordinates were plotted. To ensure that the distances between 2D points adequately reflected the Wasserstein distances among replicate distributions, we plotted the Wasserstein distances against the Euclidean distances between points (S8E and S8F Fig). The linear relationships between Wasserstein distances and Euclidean distances showed that the 2D graph accurately represents the differences between distributions.

To create the Wasserstein plot of the combined 9- and 25-dpg cell area data, cell areas of each replicate in each genotype were normalized by the mean cell area for that replicate. In this way, each replicate has a mean of 1. This eliminated the difference in the mean values of the cell size between the 9- and 25-dpg leaves, such that all higher moments of the area distributions could be compared rather than the absolute sizes.

To statistically compare the differences in the number of giant cells across genotypes in the leaf at 9 dpg and at 25 dpg, two-sample, two-tailed *t*-tests that assumed equal variance were performed on the number of giant cells per segmented area between wild type and the different genotypes.

To statistically assess the randomness of the cellular patterns, see section "Statistical analysis of the cellular patterns" below.

## Cell type classification

To automatically distinguish stomata from pavement cells, a supervised classification algorithm was used based on cell shape features (S9 Fig). Cell shape features were computed from each 2.5D mesh using the MorphoGraphX [58,59] process "Mesh/Heat Map/Analysis/Cell Analysis 2D" and were extracted with "Mesh/Attributes/Save as CSV" into a data table. Three distinct training datasets were created using a single wild-type replicate—one for the sepal, one for the leaf at 25 dpg and one for the leaf at 9 dpg. To get the different training datasets, we manually selected some pavement cells and stomata and labeled them as different cell types, ran the classification processes available within MorphoGraphX, and manually corrected the cells that were wrongly identified. These training datasets were then used to train a supervised learning algorithm (Support Vector Machine quadratic) using the Classification Learner App in MATLAB [61,62]. The following cellular shape features were selected to train the classifier in the 25-dpg leaf: area, average radius, length of the major axis, maximum radius, perimeter, circularity, lobeyness (ratio of the cell perimeter over that of its convex hull; convex shapes have lobeyness 1) [59], and rectangularity (ratio of the cell area over the area of the minimum bounding rectangle in the cell) [59]. For the sepal, the aspect ratio and the length of the minor axis were also taken into account. For the 9-dpg leaf, where the variety of cell types was more complex, three cell types were defined (pavement cell, meristemoid, and stomata) and meristemoid and stomata were combined in a postprocessing script. The shape features used to train the classifier were area, average radius, minimum radius, perimeter, circularity, lobeyness, and visibility stomata (it counts the proportion of straight lines that connects the cell outline without passing through a cell boundary) [59,63]. To automatically predict cell types in all replicates, a developed MATLAB script containing the trained classifier and a post-classifier filter, which corrects for potentially wrong predictions on the basis of known shape criteria, was applied. Manual corrections were finally performed, in which misclassified cells were re-labeled with their correct cell type.

Giant cells were defined by a cell-size threshold (S9 Fig). Because a few giant cells were expected in *atml1-3* mutants, *atml1-3* mutants were used as a reference to build this threshold. Fewer than 0.7% of the pavement cells were considered to be giant cells in *atml1-3* tissues, which was supported through visual observation in the sepal. Consequently, the giant cell-size threshold was set as that corresponding to the average between the 99.3rd percentile cell-size value with the cell-size value immediately above it in the distribution, taking into account the data of three *atml1-3* pooled replicates. For consistency, the same method was applied to the sepal, and to 9- and 25-dpg leaves, which gave three different threshold values (sepal: 5,290 $\mu m^2$, leaf 9-dpg: 2,570 $\mu m^2$, leaf 25-dpg: 14,160 $\mu m^2$). The percentiles were only calculated on rectangular sections (omitting cells at the outline of the organs) of the sepal to maintain consistency across different organs. Classification output examples in different genotypes are shown in S10 and S11 Figs.

## Randomization of the experimental images

To assess the randomness of the cellular patterns, it was essential to establish a random reference, or null model, against which the observed pattern could be compared. To produce the required random reference, the image-based method dmSET [36,37] was applied to generate 400 synthetic random equivalent tissues from each segmented image. Cell positions and orientations were randomly shuffled into new images (named randomizations) [37], while preserving individual approximate cell shapes and sizes (S15 Fig). Only the incomplete cells at the border of the images were fixed. This randomization method avoids potential biases arising from the heterogeneity of cell sizes and shapes in the tissues, which affects the number of neighboring cells. We ensured that cellular properties, and more specifically cell area and cell lobeyness (defined as the perimeter of the cell divided by the perimeter of its convex hull), were approximately conserved in the randomized leaf tissues (S15D and S15E Fig; the Pearson coefficient was 0.98 for cell area correlation and was 0.94 for cell lobeyness correlation). In the sepal randomized tissues, cell orientations were constrained between $-\pi/6$ and $+\pi/6$ compared with their initial orientation, to maintain the anisotropy of the tissue. Cell shape properties were also approximately conserved (S15I and S15J Fig). A custom-made MATLAB script was subsequently applied to both original and

randomized images to correct errors introduced by the dmSET method and to compute cell shape properties and cellular network information that was used to quantify the cellular pattern.

Before randomizing the different sepal and leaf replicate images (Figs 5 and S14), each 2.5D mesh was first converted into a 2D pixel image using the process "Stack/Mesh Interaction/Mesh To Image" (with a pixel size of 1μm) in Morpho-GraphX. Subsequently, a square crop (in the leaf) or rectangular crop (in the sepal) that maximized the tissue section was performed in the segmented images. These 2D segmented images were then randomized using the dmSET method.

To study the change in the giant cell spatial pattern over time, published time-lapse sepal data were used [39] that were randomized at two different time points (sepal at stage 4: 24 h, and at stage 8: 96 h). Cell segmentation and cell lineage tracking were already performed in [39]. Using MorphoGraphX, sepal cells were manually selected at the later time point, and the exact corresponding mother cells at the first time point were established using the lineage tracking analysis from [39]. In order to quantify the spatial pattern of the same giant cells at two different time points, giant cells at both stages were defined as the pavement cells that did not divide during this period of time. This approach allowed the comparison of the change in tissue organization consistently at two different time points. Then, the 2.5D meshes were projected into 2D images. These images were subsequently randomized using the dmSET method. Here, to facilitate the study of the same giant cells over time, the images were not cropped and the entire studied tissue was randomized, including the cells at the edges. To achieve this, the background region, located outside the tissue of interest, was considered as a single cell that remained fixed in the randomized tissues. Examples of randomizations are shown in S20C and S20D Fig. Three different sepal replicates were used for these analyses of the time-lapse data.

"Segmentation" and "Randomization" images appearing in figures such as Figs 5, 6, S14, S17, S18, and S20 were produced with a Python script using the multi-labeled images generated by dmSET.

Several approaches were used to test the robustness of our method and ensure that there was no bias in our null model. In the randomized tissues, cell sizes were well preserved (S15D Fig), but cell shapes were affected in the leaf (S15E Fig). Because the leaf pavement cells in the randomizations exhibit different shapes (more convex shapes and noisier edges) compared with the original cells, we developed an additional alternative method for generating randomized tissues (S19 Fig), called Cut and Merge Cells. In this approach, each leaf replicate was first over-segmented using Mor-phoGraphX (autoseeding with $r = 4$ μm combined with manual seeding) to create templates composed of small fragments of pavement cells (S19A Fig). These templates were then used to automatically generate a random pattern of giant cells, preserving their original sizes and numbers (S19A Fig). Giant cells were created recursively: their positions were randomly allocated to a first pavement cell fragment, which was then merged with neighboring cell fragments until the original giant cell size was reached (S19A Fig). Similar results were obtained by using this randomization method (S19B and S19C Fig), confirming the tendency of giant cells to cluster more than would be expected by chance. However, while cell shapes produced by this method exhibited less noisy edges, they remained less lobed and more convex compared with those in the original segmentation (S19F Fig). Moreover, this method does not randomize all cells in the tissue, such as stomata.

We wondered whether the shape artifacts introduced by the dmSET method could introduce bias into our null model by affecting the contacts between cells in the randomized tissues differently than in the original segmentations. To explore this possibility, we generated "reconstructions" of the original images using the same dmSET method as for the random-izations, except that each cell's position from the initial segmentation was preserved (with some added uniform noise having a maximum of ±5 μm in the $y$ and $x$ directions) (S18A–S18C, S18E, and S18F Fig). This resulted in cell shape artifacts that were nearly identical between the reconstructions and the randomizations (S18J–S18L Fig), making them more com-parable. Indeed, the differences in cell lobeyness (defined as the perimeter of the cell divided by the perimeter of its con-vex hull) from the original tissues were similar between the reconstructions and randomizations for large cells (S18J Fig). The number of cell neighbors differed slightly from the original segmentations due to the introduced shape artifacts (S18M and S18N Fig), but we found that giant cell contacts remained mostly the same in the reconstructions. Specifically, we observed approximately 6% fewer contacts between giant cells in the reconstructions compared with the segmentations.

When comparing reconstructions with randomizations (which are comparable because of their similar cell shapes), the results remained consistent (S18O Fig), with giant cells being more clustered than expected by chance ($p = 0.005$). In the sepal, cell shape was quite preserved (S15I and S15J Fig; the Pearson coefficient was >0.99 for cell area and perimeter). Sepal cell edges were noisier in the randomized tissues, with a higher cell perimeter (S15J Fig); this could potentially lead to more contacts in the randomized tissues, which would not affect our conclusion.

Furthermore, we evaluated our method by using a randomly selected population of pavement cells in both leaves and sepals (S17 Fig). The null hypothesis could not be rejected and this result was robust across five different random patterns in all replicates (one is shown in S17 Fig). Specifically, we obtained p-values ranging from 0.395 to 0.488 in leaves and from 0.178 to 0.335 in sepals. The number of giant cell contacts was either slightly more or less than the expected mean random value, attributable to variability in the random patterns. This indicates that the artificial random pattern did not deviate from true randomness, suggesting that our null models do not present a significant bias.

## Statistical analysis of the cellular patterns

By comparing a spatial observable in the cellular network of the actual segmentation with the corresponding observable in the cellular networks of the 400 generated randomized tissues, whether the considered observable is likely to be observed by chance can be statistically tested [36]. Hence, the use of this method on observables measuring distances or contacts between the studied cells allows the assessment of whether the arrangement of the cells within the tissue is random, clustered or dispersed (Fig 5A).

To quantify the patterns, a custom-made Python script was used to extract pertinent observables (i.e., spatial quantities) from the cellular network, which used the NetworkX Python library [64]. In this manuscript, we mainly focused on the number of giant cell neighbors per giant cell to quantify the number of local contacts between giant cells. Other observables have been quantified, such as the minimum shortest path between giant cells, and the number of giant cells in a cluster (S16 Fig). When dealing with cropped images, giant cells (or any cell population studied, see Fig 6) at the border of the image were not considered in the analysis.

The number of giant cell neighbors was extracted for every giant cell, and the mean number of giant cell neighbors per giant cell across all giant cells was computed within each experimental replicate. Similarly to the methodology described by the authors of the dmSET method [36], the mean value extracted from the segmentation image was compared with the approximated null distribution formed by the 400 mean values extracted from the randomized images. We first performed the analysis on each replicate independently (S21 and S22 Figs). As the cell-size distributions in the different replicates showed similarities across replicates (Figs 5, S21, and S22), replicates were pooled to increase the sample size and statistical power. Six image replicates were used for both the leaf and sepal wild-type (Fig 5), and three replicates were used for the wild-type sepal time-lapsed images (Fig 7). To test the null hypothesis assessing the randomness of the observed metric, a p-value $p$ was obtained as the ratio of the number of random images (defined here as one random image resulting from pooling one random image per replicate) displaying the same or a more extreme value than the one obtained in the segmentation replicates (one-sided test). If the value fell within the null distribution with an associated high p-value ($p > 0.05$), the null hypothesis could not be rejected, indicating that the observed quantity could likely be expected by chance. If the value fell outside the null distribution, we assigned $p < 0.0025$, with 0.0025 corresponding to the inverse of the number of random images (400) used to create the null distribution.

In addition, the distribution of the number of giant cell neighbors for all giant cells from the pooled experimental replicates was studied, which provided more insights into their spatial organization. This was compared qualitatively with the distribution expected in a random tissue, extracted from the 400 randomized tissues of all replicates.

All plots derived from these analyses were performed with Python, with the use of the matplotlib [65] and seaborn packages [66].

## Mathematical model for giant cell fate commitment and numerical simulations

To simulate the giant cell fate decisions, our published stochastic and cell-autonomous multicellular model in a growing tissue was used [12]. In that model, the transcription factor ATML1 stochastically fluctuates and drives the expression of its target LGO. In the simulated growing tissue, cells divide using a timer with some stochasticity. When the timer of a cell reaches a threshold $\Theta_{C,S}$, cells undergo the S-phase, and therefore cells transition from being diploid (2C) to tetraploid (4C). By default, cells that reach a second and higher timer threshold $\Theta_{C,D}$ will undergo division. However, those cells that have reached a certain LGO concentration threshold $\Theta_T$ after undergoing the S-phase, considered to be in the G2 phase, will not divide and are maintained in an endoreduplication cycle, which increases their ploidy, becoming giant cells.

To model the dynamics of the concentrations for ATML1 and LGO, and the Timer variable, we use chemical Langevin equations [67], which are differential equations with a corresponding deterministic part, consisting of production, degradation and regulatory terms, followed by a stochastic part modeling thermodynamic fluctuations that contains a square root, whose radicand has the sum of the absolute values of the production, degradation, and regulatory terms. The dynamics of ATML1 concentration, LGO concentration and Timer variable in cell $i$, denoted by $[ATML1]_i$, $[Target]_i$ and $Timer_i$, respectively, follow the Langevin equations given by

$$\frac{d[ATML1]_i}{dt} = P_A + \frac{V_A[ATML1]_i^{n_A}}{K_A^{n_A} + [ATML1]_i^{n_A}} - G_A[ATML1]_i + \sqrt{\frac{1}{2\varepsilon_i(t)}\left(P_A + \frac{V_A[ATML1]_i^{n_A}}{K_A^{n_A} + [ATML1]_i^{n_A}} + G_A[ATML1]_i\right)}\eta_{ATML1,i} \tag{1}$$

$$\frac{d[Target]_i}{dt} = \frac{V_T[ATML1]_i^{n_T}}{K_T^{n_T} + [ATML1]_i^{n_T}} - G_T[Target]_i + \sqrt{\frac{1}{2\varepsilon_i(t)}\left(\frac{V_T[ATML1]_i^{n_T}}{K_T^{n_T} + [ATML1]_i^{n_T}} + G_T[Target]_i\right)}\eta_{Target,i} \tag{2}$$

$$\frac{dTimer_i}{dt} = P_C + \sqrt{\frac{1}{2\varepsilon_i(t)}(P_C)}\eta_{Timer,i}. \tag{3}$$

Eqn. (1) stands for the rate of change of ATML1 in cell $i$, and its terms on the right-hand side describe constitutive expression, self-activation (implemented via a Hill function) [68], linear degradation, and the corresponding stochastic term in ATML1; Eqn. (2), stands for the rate of change of the Target in cell $i$, and its terms on the right-hand side describe ATML1-induced expression of the Target (using also a Hill function), linear degradation, and the corresponding stochastic term in the Target; Eqn. (3) stands for the rate of change of the Timer in cell $i$, and its terms on the right-hand side are a constitutive production and the corresponding stochastic term. The parameters of the equations are as follows: $P_X$ is the basal production rate for the $X$ variable (where $X$ is either $A$ for ATML1, $T$ for Target concentration or $C$ for the Timer variable), $V_X$ is the prefactor of the ATML1-dependent production rate for the $X$ variable, $K_X$ is the ATML1 concentration at which the ATML1-dependent production rate for the variable $X$ has its half-maximal value, $n_X$ is the Hill coefficient, and $G_X$ is the linear degradation rate for the $X$ variable. $\varepsilon_i(t)$ is a normalized cell area, $\varepsilon_i(t) = E_0E_i(t)$, where $E_0$ is an effective cell area, and $E_i(t)$ is the area of cell $i$ in arbitrary units. $\eta_{Xi}$ is a random Gaussian variable with zero mean that fulfills $\langle\eta_{Xi}(t)\eta_{X'j}(t')\rangle = \delta(t-t')\delta_{XX'}\delta_{ij}$, where $i$ and $j$ are cell indices, $X$ and $X'$ the modeled variables, $\delta_{XX'}$ and $\delta_{ij}$ are Kronecker deltas and $\delta(t-t')$ is the Dirac delta function.

Upon cell division, the Timer was reset. To implement the resetting, the following rule was applied at each time step:

$$Timer_i(t) \rightarrow \{U_i \text{ if } Timer_i(t) \geq \Theta_{C,D};\ Timer_i(t)\quad \text{otherwise}\}, \tag{4}$$

where $U_i$ is a uniform randomly distributed number in the interval [0, 0.5) and $\Theta_{C,D}$ is the cell division threshold for the Timer.

The multicellular template on which the simulations were run and initial conditions were the same as in Meyer and colleagues (2017). Initial conditions for ATML1 and Target were randomly uniformly distributed in the interval of [0,1) and [0,0.1), respectively. The Timer initial conditions were set in correlation with the cell areas in the initial template with some stochasticity, as performed in Meyer and colleagues (2017). The differences between the used initial conditions were just in the ATML1, Target, and Timer initial cellular values, determined by different random numbers.

Tissue growth and division were also implemented as in Meyer and colleagues (2017). The multicellular tissue grows anisotropically, to emulate the patterning process in the sepal. This anisotropic growth was implemented by imposing a displacement of the vertices with respect to the center of mass of the tissue, with a given radial and vertical exponential rate. After each simulation step, dilution effects due to growth in the modeled variables were taken into account. Cells divided using the shortest path rule together with the constraint of having the division plane through the center of mass of the cell. We assumed that molecules are homogeneously distributed within cells, and therefore, upon cell division, sister cells have the same concentration of ATML1 and Target variables at birth, but can have different cell sizes.

Numerical simulations were performed with Tissue software [12,13,69], and the integration was performed using an Îto interpretation of the Langevin equations with a Heun algorithm [70]. Integration was performed with a time step $dt = 0.1$, and simulations were stopped at time 135. Parameter values for the simulations are given in S2 Table. Parameters were chosen such that the wild-type behavior reported in Meyer and colleagues (2017) could be recapitulated. The outcome of the simulation in Fig 7B was displayed using Paraview software [71].

We recently proposed a more detailed model of the ATML1 regulatory network to study how giant cell specification and cell fate maintenance depend on VLCFA [13], which is still a stochastic and cell-autonomous model. Here, however, for the sake of simplicity, and the intention of using a minimal, stochastic, and cell-autonomous phenomenological model, the former ATML1 model was used [12].

## Randomizations of the outcomes from the numerical simulations

To assess the randomness of the giant cell pattern in the numerical simulations (Fig 7), the same method was employed as that used for the experimental images. Although randomizations of the tissues were performed similarly (see the "Randomization of the experimental images" section above), a Python script was developed to display the output of the simulation as a multi-labeled image, where each cell was colored with a different label. These images could therefore be randomized using the dmSET method [36,37]. To compare the simulated giant cell pattern (Fig 7B) with the giant cell pattern found in the experimental mature sepals (Fig 5B), the output image was cropped using the maximal rectangle in the tissue, and giant cells were also defined by a size threshold, ensuring that all cells with a ploidy of 16C or higher were categorized as giant cells (Fig 7C). The few 8C cells that exceeded this threshold were also considered as giant cells.

To study the change in the giant cell spatial pattern over time (Fig 7F), the same simulations were used, but only the first-arising giant cells (i.e., cells that stopped dividing after time $t = 55$ of the simulations for being committed to endoreduplicate) were studied. The same method was used to assess the randomness of the cellular patterns on these giant cells both at time $t = 55$ and time $t = 135$. Here, instead of cropping the image, the whole tissue was randomized (using the dmSET method), including the cells at the edges, such that exactly the same giant cells were considered at both time points. Examples of randomizations are shown in S20A and S20B Fig. The analysis was performed over five simulation replicates, with different cellular random initial conditions.

Related "Segmentation" and "Randomization" images appearing in Figs 7 and S20 were produced with a Python script using the multi-labeled images generated by dmSET.

## Supporting information

**S1 Fig. Cauline leaves have elongated giant cells similar to sepals.** (**A**) Abaxial side of a wild-type mature sepal expressing a cell membrane marker (*p35S::mCitrine-RCI2A*). (**B**) Tip section of the abaxial side of a wild-type cauline leaf expressing a cell membrane marker (*p35S::mCitrine-RCI2A*). (**C**) A developing abaxial side of a wild-type rosette leaf 1 or 2 at 8 dpg expressing a cell membrane marker (*p35S::mCitrine-RCI2A*). Scale bars associated with the overview images (bottom) represent 200 μm, and scale bars associated with the magnified images represent 100 μm. The underlying data for this figure can be found at Open Science Framework (osf.io), https://doi.org/10.17605/OSF.IO/RFCWS.
(TIF)

**S2 Fig. Replicates of abaxial and adaxial cell-size distribution in the wild-type leaf and sepal epidermis; cell size correlates with DNA content.** Cell area heat maps (in μm$^2$) of (**A, C**) abaxial surfaces of wild-type sepals, (**B, D**) adaxial surfaces of wild-type sepals, (**E**) abaxial surface of 25-dpg wild-type leaf 1 or 2 (cell density: 284 cells mm$^{-2}$) and (**F**) adaxial surface of 25-dpg wild-type leaf 1 or 2 (cell density: 177 cells mm$^{-2}$). Scale bars represent 100 μm. (**G**) Abaxial and adaxial side of 25-dpg leaf cell area versus DNA content (one of two replicates) as measured by H2B-TFP total nuclear fluorescence, with $R^2 = 0.91$ for the abaxial side and $R^2 = 0.79$ for the adaxial side. Associated with Fig 2. (**H**) Cell area of the largest cells (area >4,308 μm$^2$) versus DNA content as measured by H2B-TFP total nuclear fluorescence in both the abaxial and adaxial side of the 25-dpg leaf (red) and in the abaxial side of the adult sepal (blue). The underlying data for this figure can be found at Open Science Framework (osf.io), https://doi.org/10.17605/OSF.IO/RFCWS.
(TIF)

**S3 Fig. Cell-size patterning occurs as a basipetal wave simultaneously in the adaxial and abaxial sides of the leaf.** (**A–B**) Cell area heat maps (in μm$^2$) for wild-type leaf 1 or 2 leaves at 5–9 dpg on (**A**) the abaxial side of the leaf and on (**B**) the adaxial side of the same leaf. Scale bars represent 100 μm. Leaves are to scale and have the same heat map color range. Associated with Fig 3. The underlying data for this figure can be found at Open Science Framework (osf.io), https://doi.org/10.17605/OSF.IO/RFCWS.
(TIF)

**S4 Fig. The sepal giant cell specification pathway also patterns cell size in 9-dpg leaves, replicate 2.** Cell area heat maps (in μm$^2$) of the upper abaxial quadrant of leaf 1 or 2 at 9 dpg for the genotypes (**A**) wild type, (**B**) *acr4-2*, (**C**) *atml1-3*, (**D**) *dek1-4*, (**E**) *lgo-2*, (**F**) *LGO-OX* (*pATML1::LGO*), and (**G**) *ATML1-OX* (*pPDF1::ATML1*). Scale bar represents 100 μm. Second replicate associated with Fig 4. (**H**) 2D Wasserstein distance plot for 9-dpg replicates. Cell area heat maps of other replicates are shown in Figs 4 and S5. The Wasserstein statistical tests among replicates are shown in S8 Fig. The underlying data for this figure can be found at Open Science Framework (osf.io), https://doi.org/10.17605/OSF.IO/RFCWS.
(TIF)

**S5 Fig. The sepal giant cell specification pathway also patterns cell size in 9-dpg leaves, replicate 3.** Cell area heat maps (in μm$^2$) of the upper abaxial quadrant of leaf 1 or 2 at 9 dpg for the genotypes (**A**) wild type, (**B**) *acr4-2*, (**C**) *atml1-3*, (**D**) *dek1-4*, (**E**) *lgo-2*, (**F**) *LGO-OX* (*pATML1::LGO*), and (**G**) *ATML1-OX* (*pPDF1::ATML1*). Scale bar represents 100 μm. Third replicate associated with Fig 4. The underlying data for this figure can be found at Open Science Framework (osf.io), https://doi.org/10.17605/OSF.IO/RFCWS.
(TIF)

**S6 Fig. The sepal giant cell specification pathway also patterns cell size in 25-dpg mature leaves, replicate 2.** Cell area heat maps (in μm$^2$) of a region approximately midway between midrib and margin and between tip and base on the abaxial side of leaf 1 or 2 at 25 dpg for the genotypes (**A**) wild type, (**B**) *acr4-2*, (**C**) *atml1-3*, (**D**) *dek1-4*, (**E**) *lgo-2*, (**F**)

*LGO-OX* (*pATML1::LGO*), and (**G**) *ATML1-OX* (*pPDF1::ATML1*). Scale bar represents 100 µm. Second replicate associated with Fig 4. (**H**) Wasserstein distances for 25-dpg replicates displayed as Euclidean distances embedded in 2D. Cell area heat maps of other replicates are shown in Figs 4 and S7. Datasets from (E) and (F) are also used for an independent analysis in Trozzi and colleagues (2023). The underlying data for this figure can be found at Open Science Framework (osf.io), https://doi.org/10.17605/OSF.IO/RFCWS.
(TIF)

**S7 Fig. The sepal giant cell specification pathway also patterns cell size in 25-dpg mature leaves, replicate 3.** Cell area heat maps (in µm$^2$) of an area approximately midway between midrib and margin and between tip and base on the abaxial side of leaf 1 or 2 at 25 dpg for the genotypes (**A, B**) wild type (two replicates), (**C**) *acr4-2*, (**D**) *atml1-3*, (**E**) *dek1-4*, (**F**) *lgo-2*, (**G**) *LGO-OX* (*pATML1::LGO*), and (**H**) *ATML1-OX* (*pPDF1::ATML1*). Scale bar is 100 µm. Third replicate associated with Fig 4. Datasets from (A), (B), (F), and (G) are also used for an independent analysis in Trozzi and colleagues (2023). The underlying data for this figure can be found at Open Science Framework (osf.io), https://doi.org/10.17605/OSF.IO/RFCWS.
(TIF)

**S8 Fig. Statistical tests on the cell-size distributions and statistical tests in young and mature leaves.** (**A, B**) Violin plots of cell area densities on a log$_{10}$ scale for individual replicates of (A) 9-dpg and (B) 25-dpg leaves. Stomata were removed in both (A) and (B). Associated with Fig 4C and 4D. (**C, D**) *p*-values of the Wasserstein tests for all the replicate pair comparisons for (C) 9-dpg and (D) 25-dpg leaves. (**E, F**) Wasserstein test statistics plotted against their corresponding Euclidean distances for all (E) 9-dpg and (F) 25-dpg test pairs. Associated with Fig 4. The underlying data for this figure can be found at Open Science Framework (osf.io), https://doi.org/10.17605/OSF.IO/RFCWS.
(TIF)

**S9 Fig. Classification of cell types in the leaf and the sepal.** (**A–F**) Steps for cell-type classification, with a wild-type sepal (A–C) and a wild-type leaf 25 dpg (D–F) as examples. First, cellular shape features were computed from the segmented meshes in MorphoGraphX (A, D) to train a classification algorithm (Support Vector Machine [SVM] quadratic) and predict the stomata (blue) and the pavement (green) cell types (B, E). Heat map colors represent the cell area in (A, D). Second, to classify giant cells (C, F), a size threshold was established based on observations in *atml1-3* sepals, such as a small fraction of pavement cells (0.7% on average) are giant cells in *atml1-3* mutants (see G, H). Heat map colors represent cell type in (B, C, E, F). (**G, H**) Cell size in three wild-type replicates and the three *atml1-3* replicates that were used to construct the threshold in leaf 25 dpg (G) and in the sepal (H). Points are colored according to the cell-type classification. Cells exceeding the defined size threshold (see Materials and methods) are defined as giant cells (magenta). Associated with S10 Fig. The code and data associated with this figure can be found at Open Science Framework (osf.io), https://doi.org/10.17605/OSF.IO/RFCWS.
(TIF)

**S10 Fig. Cell type classification in the leaf and sepal.** (**A, B**) Segmented meshes of one replicate for each genotype after cell type classification in the mature sepal (A) and the mature leaf (B). Cells are colored with their corresponding cell type: pavement cells (in green), stomata (in blue) and giant cells (in magenta). Stomata and pavement cells were first classified using a trained classification algorithm based on cell shape features. Giant cells were defined as the largest cells, using a size threshold based on *atml1-3* mutants (see Materials and methods). Scale bars represent 200 µm. See also S9 and S11 Figs. The code and data associated with this figure can be found at Open Science Framework (osf.io), https://doi.org/10.17605/OSF.IO/RFCWS.
(TIF)

**S11 Fig. Cell type classification in the 9-dpg leaves.** Meshes of one replicate for each genotype in the 9-dpg leaf sections. Cells are colored with their corresponding cell type: pavement cells (in green), stomata (in blue) and giant cells (in magenta). Stomata and pavement cells were first classified using a trained SVM classification algorithm based on cell shape features. Because meristemoids and stomata are difficult to distinguish at this stage, both were classified as stomata. Giant cells were defined by a cell-size threshold based on *atml1-3* mutants (see Materials and methods and S9 Fig). Scale bar represents 200 µm. Associated with S10 Fig. The code and data associated with this figure can be found at Open Science Framework (osf.io), https://doi.org/10.17605/OSF.IO/RFCWS. (TIF)

**S12 Fig. Cell-size patterning has little effect on leaf size, except in *ATML1-OX* and *LGO-OX*.** (A–G) Images of 9 dpg full leaves with *p35S::mCitrine-RCI2A* used for segmenting the upper quadrant (Fig 4). One replicate of each genotype is shown. Leaves of similar sizes were chosen to developmentally stage match as closely as possible. (H) Leaf areas of three replicates of 9-dpg leaves for each genotype. (I–O) Images of 25 dpg full leaves with *p35S::mCitrine-RCI2A* used for imaging and segmenting the mature cells (Fig 4). One replicate of each genotype is shown. (P) Leaf areas of three replicates of 25-dpg leaves for each genotype. Scale bar is 0.5 mm for (A–G) and 1 mm for (I–O). The underlying data for this figure can be found at Open Science Framework (osf.io), https://doi.org/10.17605/OSF.IO/RFCWS. (TIF)

**S13 Fig. Giant cells are not preferentially positioned overlying the vasculature.** (A) Top: Cell area heat map of the abaxial upper quadrant of a 9-dpg wild-type leaf. Large cells overlying the midrib that extend up to the leaf tip are boxed in white. Large cells that align as if along an underlying vein are circled in white. Bottom: Cell area heat map of the abaxial midrib region of a 9-dpg wild-type leaf. A large pavement cell extending out from a large midrib cell as if following an underlying vein peeling off the midrib. (B) Cell area heat map of the abaxial side of half of a 9-dpg wild-type leaf with the underlying vasculature in white. The colored heat map is associated with the color bar in (A). (C–F) Cell area heat map of the abaxial sides of halves of *ATML1-OX* leaves with the underlying vasculature in white (four replicates; see Materials and methods). Colored heat maps are associated with the color bar in (F). All scale bars represent 100 µm. The underlying data for this figure can be found at Open Science Framework (osf.io), https://doi.org/10.17605/OSF.IO/RFCWS. (TIF)

**S14 Fig. Examples of randomizations in the leaf and in the sepal.** (A) One replicate of the leaf after cell type classification and three corresponding randomized tissues as an example. (B) One replicate of the sepal after cell type classification and three corresponding randomized tissues as an example. Associated with Fig 5. The code and data associated with this figure can be found at Open Science Framework (osf.io), https://doi.org/10.17605/OSF.IO/RFCWS. (TIF)

**S15 Fig. Conservation of cellular features between the segmented tissue and the corresponding randomized tissues.** (A) Example of a segmented tissue and a corresponding randomized tissue, generated using the dmSET method, in the leaf 25 dpg (the replicate used is the same as in Fig 5B). (B, C) Comparison of the cell center coordinates (in µm) between the real tissue (segmentation) and a randomized tissue (dmSET randomization) shown in (A). (D, E) Comparison of cell shape features between all real tissues (segmentation) and one of their randomized tissues (dmSET randomization): (D) cell area (in µm$^2$) and (E) cell lobeyness (defined as the perimeter of the cell divided by the perimeter of its convex hull). (F) Example of a segmented tissue and a corresponding randomized tissue, generated using the dmSET method, in the sepal (the replicate used is the same as in Fig 5B). (G, H) Comparison of the cell center coordinates (in µm) between the real tissue (segmentation) and a randomized tissue (dmSET randomization) shown in (F). (I, J) cell shape features between all real tissues (segmentation) and one of their randomized tissues (dmSET randomization): cell area (I) and cell perimeter (J). The color bars associated with (B–C), (D–E), (G–H), and (I–J) represent the cell areas (in

$\mu m^2$). Each dot represents one cell. Associated with Fig 5. The code and data associated with this figure can be found at Open Science Framework (osf.io), https://doi.org/10.17605/OSF.IO/RFCWS. (TIF)

**S16 Fig. Different spatial observables were studied to statistically assess the spatial organization of giant cells.** Three measures were extracted from the cellular network of the segmentations (red) and statistically compared against the randomizations (gray). (**A–D**) The number of giant cell neighbors per giant, as a mean over giant cells (A, C) and as a distribution (B, D), as presented in Fig 5C and 5D, respectively. (**E–H**) The minimum shortest path (min. SP) between two giant cells (in path length, 1 meaning that the pair of giant cells are in contact) as a mean over giant cells (E, G) and as a distribution (F, H). (**I, K**) Fraction of giant cells in contact and (**J, L**) number of giant cells per giant cell cluster as a distribution. The top panels (A, B, E, F, I, J) show the result of the pattern quantification in the wild-type leaf 25 dpg and the bottom panels (C, D, G, H, K, L) show the results in the wild-type sepal. Analyses were performed over six pooled replicates. Total number of giant cells counted (excluding giant cells at the image border) in the analysis: $n = 68$ (leaf, segmentations), $n = 68 \times 400$ (leaf, randomizations), $n = 74$ (sepal, segmentations), $n = 74 \times 400$ (sepal, randomizations). The code and data associated with this figure can be found at Open Science Framework (osf.io), https://doi.org/10.17605/OSF.IO/RFCWS. (TIF)

**S17 Fig. Assessment of the dmSET method on a random cell pattern.** (**A, B**) To identify potential biases in our null model due to artifacts in randomized tissues, an artificial random cellular pattern was created in all leaf replicates by randomly selecting a population of pavement cells with an area larger than 2,000 $\mu m^2$ in leaves. (A) Example of representative segmentation of a 25-dpg wild-type leaf (left) and one of its corresponding randomized tissue (right). Randomly selected cells are labeled in magenta. (B) Mean number of cell neighbors per cell within the randomly selected cells in the real tissues (segmentations) and in the randomized tissues (randomizations). The null hypothesis could not be rejected, showing that the artificial random pattern does not deviate significantly from randomness (see Materials and methods). (**C, D**) Similar to (A–B) in the sepals by randomly selecting a population of pavement cells with an area larger than 100 $\mu m^2$. Total number of cells considered in the analysis: $n = 10 \times 6$ (leaf, segmentations), $n = 10 \times 6 \times 400$ (leaf, randomizations), $n = 30 \times 6$ (sepal, segmentations), $n = 30 \times 6 \times 400$ (sepal, randomizations). The code and data associated with this figure can be found at Open Science Framework (osf.io), https://doi.org/10.17605/OSF.IO/RFCWS. (TIF)

**S18 Fig. Reconstruction of the tissues using dmSET to investigate the effects of shape artifacts in randomized tissues.** (**A–C**) Example of (**A**) an original segmentation of a 25-dpg wild-type leaf, (**B**) a reconstruction, and (**C**) one randomized tissue. All leaf replicates were reconstructed as shown in (C) by using the same dmSET used for the generation of randomizations (see Materials and methods), but constraining each cell's position to its original location plus a small amount of noise. (**D**) Average cell neighboring areas versus cell areas in real tissues and in randomizations. (**E**) Cell positions (in $\mu m$) in the reconstruction of all replicates are nearly equal to segmentations. (**F**) Cell positions (in $\mu m$) in one randomization of all replicates are randomly shuffled. (**G**) Relative error on cell areas between the segmentation (or reconstruction) and the randomizations. (**H**) Cell areas (in $\mu m^2$) in reconstructions versus segmentation. (**I**) Cell areas (in $\mu m^2$) in randomizations versus segmentation. (G–I) show that cell areas are similar in segmentations, reconstructions and in the randomized tissues. (**J**) Relative error on cell lobeyness between the segmentation (or reconstruction) and the randomizations. (**K**) Cell lobeyness in reconstructions versus segmentation. (**L**) Cell lobeyness in randomizations versus segmentation. The color bar in (E) denoting cell areas is also associated with panels (F, H, I, K, L, and N). (J–L) show that cell shapes are affected in a similar manner in reconstructions and original segmentations for cells larger than 10,000 $\mu m^2$. (**M**) Number of neighbors as a function of cell areas in reconstructions and in segmentations are similar. (**N**) Number of neighbors in reconstructions versus segmentation. (**O**) Mean number of giant cell neighbors per giant cell (as shown

in Fig 5C) with the comparison with the tissues "reconstructions". (M–O) show that the cell connectivity is affected in the reconstruction due to the shape artifacts, but that giant cell contacts are not significantly different and the results remain the same when comparing the reconstructions with the randomizations (see Materials and methods). The six replicates were compared together with one corresponding randomized tissue in (D–N). $\rho$ indicates Pearson correlation coefficient in (E, F, H, I, K, L, and N). Two-sample $t$ tests were performed in (G, J, M) at each interval: $p < 0.05$ (*), $p < 0.01$ (**), $p < 0.001$ (***). The code and data associated with this figure can be found at Open Science Framework (osf.io), https://doi.org/10.17605/OSF.IO/RFCWS.
(TIF)

**S19 Fig. Giant cells are more clustered than in a randomized null model using the Cut and Merge Cells method.** (**A**) Overview of the alternative method referred to as Cut and Merge Cells (CMC) to generate random giant cell patterns (see Materials and methods). Each leaf replicate was over-segmented to create templates made of small pieces of pavement cells. These templates were then used to automatically generate a random pattern of giant cells by preserving their sizes and numbers (see Materials and methods). (**B**) Example of one initial segmentation of a 25-dpg leaf (left) and one corresponding CMC randomization (right). (**C, D**) Quantification of the giant cell patterns to compare with Fig 5C and 5D. (C) Mean number of giant cell neighbors per giant cell in the real tissues (segmentation) and in their randomizations (in gray). (D) Distributions of the number of giant cell neighbors for all giant cells found in all replicates of segmentations (in red) and randomizations (in gray). (**E**) Left: Comparison of cell areas of giant cells in the dmSET randomizations and in the segmentations. Right: Comparison of cell areas of giant cells in the CMC randomizations and in the segmentations. (**F**) Left: Comparison of cell lobeyness (see Materials and methods) of giant cells in the dmSET randomizations and in the segmentations. Right: Comparison of cell lobeyness of giant cells in the CMC randomizations and in the segmentations. Six replicates and one randomization per replicate were considered in (E, F). The color bars in (E, F) denote cell areas in μm². The code and data associated with this figure can be found at Open Science Framework (osf.io), https://doi.org/10.17605/OSF.IO/RFCWS.
(TIF)

**S20 Fig. Examples of randomizations at initial and final time points in simulations and in time-lapse imaging sepal data.** One replicate is shown on the left and three corresponding randomized tissues on the right as an example. (**A, B**) Numerical simulation at time $t = 55$ (initial time) (A) and at time $t = 135$ (final time) (B). (**C, D**) Sepal from time-lapse imaging data at two different stages called here initial time (C) and final time (D). Associated with Fig 7. The real sepals in (C) and (D) were also used for an independent analysis in Hervieux and colleagues (2016). The code and data associated with this figure can be found at Open Science Framework (osf.io), https://doi.org/10.17605/OSF.IO/RFCWS.
(TIF)

**S21 Fig. Quantification of the giant cell pattern in the individual replicates of the leaf.** On the left, each replicate is displayed after the cell type classification. In the middle: the corresponding results of the cellular pattern quantification showing the mean giant number of neighbors per giant, where the tissue segmentations (in red) were statistically compared with the tissue randomizations, containing 400 randomized tissues per replicate (in gray). On the right: the corresponding distributions of the number of giant cell neighbors per giant cell, both for the actual tissue (red) and the randomizations (gray). Associated with Fig 5. The code and data associated with this figure can be found at Open Science Framework (osf.io), https://doi.org/10.17605/OSF.IO/RFCWS.
(TIF)

**S22 Fig. Quantification of the giant cell pattern in the individual replicates of the sepal.** On the left, each replicate is displayed (rotated 90°) after the cell type classification. In the middle: the corresponding results of the cellular pattern quantification showing the mean giant number of neighbors per giant, where the tissue segmentations (in red) are

statistically compared with the randomizations, containing 400 randomized tissues per replicate (in gray). On the right: the corresponding distributions of the number of giant cell neighbors per giant cell, both for the actual tissue (red) and the randomizations (gray). Associated with Fig 5. The code and data associated with this figure can be found at Open Science Framework (osf.io), https://doi.org/10.17605/OSF.IO/RFCWS.
(TIF)

**S1 Table. Primers for genotyping mutants and overexpression lines.**
(PDF)

**S2 Table. Parameter values used for the simulations.** Parameters used for the simulations shown in Figs 7 and S20. All units are arbitrary.
(PDF)

## Acknowledgments

We thank Nicholas J. Russell, John Chandler, Josep Mercadal, Elise Laruelle, and Philippe Andrey for critical comments on the manuscript. We thank Stephen Starkman, Emily Phung, and Isabel Delo for assistance with segmentation, and Violeta Gibelli for assistance with cell classification corrections. We thank Richard Smith and Soeren Strauss for their help in using MorphoGraphX; we are grateful that Richard Smith created a process to generate 2D images from meshes. We also thank Philippe Andrey for his insights on image analysis and statistics and Elise Laruelle for her help in using the dmSET randomization code, as well as the Cornell Statistics Consulting Unit, specifically May Boggess, for her help with statistics and coding.

## Author contributions

**Conceptualization:** Frances K. Clark, Gauthier Weissbart, Pau Formosa-Jordan, Adrienne H. K. Roeder.

**Data curation:** Gauthier Weissbart, Frances K. Clark, Xihang Wang, Adrienne H. K. Roeder, Pau Formosa-Jordan.

**Formal analysis:** Gauthier Weissbart, Frances K. Clark, Chun-Biu Li, Xihang Wang.

**Funding acquisition:** Pau Formosa-Jordan, Adrienne H. K. Roeder.

**Investigation:** Frances K. Clark, Gauthier Weissbart, Xihang Wang.

**Project administration:** Pau Formosa-Jordan, Adrienne H. K. Roeder.

**Resources:** Kate Harline.

**Software:** Gauthier Weissbart, Pau Formosa-Jordan.

**Supervision:** Pau Formosa-Jordan, Adrienne H. K. Roeder.

**Visualization:** Gauthier Weissbart, Frances K. Clark.

**Writing – original draft:** Frances K. Clark, Gauthier Weissbart, Pau Formosa-Jordan, Adrienne H. K. Roeder.

**Writing – review & editing:** Gauthier Weissbart, Frances K. Clark, Chun-Biu Li, Xihang Wang, Pau Formosa-Jordan, Adrienne H. K. Roeder.

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
