## [Editor Report · Decision Letter 0]

21 Nov 2024

Dear Dr Roeder,

Thank you for submitting your manuscript entitled "A common pathway controls cell size in the sepal and leaf epidermis leading to a non-random pattern of giant cells" for consideration as a Research Article by PLOS Biology.

Your manuscript has now been evaluated by the PLOS Biology editorial staff as well as by an academic editor with relevant expertise and I am writing to let you know that we would like to send your submission out for external peer review.

Once your full submission is complete, your paper will undergo a series of checks in preparation for peer review. After your manuscript has passed the checks it will be sent out for review. To provide the metadata for your submission, please Login to Editorial Manager (https://www.editorialmanager.com/pbiology) within two working days, i.e. by Nov 25 2024 11:59PM.

Kind regards,

Ines

--

Ines Alvarez-Garcia, PhD

Senior Editor

PLOS Biology

---

## [Decision Letter · Decision Letter 1]

9 Jan 2025

Dear Dr Roeder,

Thank you for your patience while your manuscript entitled "A common pathway controls cell size in the sepal and leaf epidermis leading to a non-random pattern of giant cells" was peer-reviewed at PLOS Biology. Please also accept my apologies for the delay in sending you our decision. Your manuscript has been evaluated by the PLOS Biology editors, an Academic Editor with relevant expertise, and by two independent reviewers.

The reviewers are attached below. As you will see, the reviewers find the paper beautifully done and very informative for researchers working in the field, however they also raise several issues. Reviewer 1 is not convinced of the overall advance over previous publications, but also thinks that the description of the cell size and shape throughout development on both sides of the leaf is interesting for the field. Reviewer 2 raises issues with the presentation, as the paper is challenging to read, and makes several suggestions to improve it and avoid repetition of the data and better comparisons. This reviewer also mentions that the randomisation method used is not clear and is not convinced that it works for pavement cells as it can introduce artifacts into the biological structure of the null models, and suggests a different method that removes shared sections and introduces new division planes. In addition, the reviewer thinks that instead of the cells getting more clustered, as proposed, the order they remember from the simpler conditions could get more and more different with respect to the increasing entropy of the tissue as a whole. You should consider this interpretation of the results.

Based on the specific reviewers' comments and following discussion with the Academic Editor, it is clear that a substantial amount of work would be required to meet the criteria for publication in PLOS Biology. However, given our and the reviewers interest in your study, we would be open to inviting a comprehensive revision of the study that thoroughly addresses all the reviewers' comments. Given the extent of revision that would be needed, we cannot make a decision about publication until we have seen the revised manuscript and your response to the reviewers' comments. Your revised manuscript would need to be seen by the reviewers again, but please note that we would not engage them unless their main concerns have been addressed.

We appreciate that these requests represent a great deal of extra work, and we are willing to relax our standard revision time to allow you 6 months to revise your study. Please email us (plosbiology@plos.org) if you have any questions or concerns, or envision needing a (short) extension. At this stage, your manuscript remains formally under active consideration at our journal; please notify us by email if you do not intend to submit a revision so that we may withdraw it.

**IMPORTANT - SUBMITTING YOUR REVISION**

3. Resubmission Checklist

a) *PLOS Data Policy*

b) *Published Peer Review*

Sincerely,

Ines

--

Ines Alvarez-Garcia, PhD

Senior Editor

PLOS Biology

Reviewers' comments

Rev. 1:

In this paper, the authors analyse the distribution of giant cells in arabidopsis leaves and show that the same genetic pathway controls giant cell formation in both leaves and sepals. They also generate a computational model that explains how initially randomly distributed giant cell precursors transition to a clustered, non-random distribution.

The work is carefully done and clearly explained, with beautiful images of leaves beautifully quantified. The authors have also gone to considerable effort to quantify development on the upper and lower surfaces of the leaf. This is rarely done, and gives a much fuller view of leaf patterning. As a description of epidermal cell patterning in leaf development, it is outstanding. However, the scientific advance (that the same genetic pathway regulates giant cells in the leaf as well as the sepal) seems fairly minor.

It is already known that lgo mutants have altered pavement cell division, size and endoreduplication in leaves and that LGO's paralogues in the SIM/SMR family alter pavement cell size (Roeder et al. 2010, Churchman et al. 2006, Kumar et al. 2015 and Dubois et al. 2023). The model presented here is essentially the same as one already published by the authors and is known to explain the distribution of giant cells in sepals (Meyer et al. 2017). Given this previous data, the finding that the same pathway works to regulate giant cells in leaves as well as sepals seems unsurprising. The beautiful description of cell size and shape throughout development on both sides of the leaf will be of interest to other researchers in the field though.

Rev. 2: Iain Johnston – note that this reviewer has signed his review.

This is a detailed survey of an interesting and deep aspect of plant biology – the control of physical properties of cell size and positioning. Some beautiful microscopy and comprehensive mutant analysis paints a picture connecting the same circuitry to cell size control in two di<erent cases. Computational and modelling theory is developed to explore evidence for cell properties departing from random null models, and to explore a putative explanatory mechanism for the links observed.

I enjoyed reading this paper and think its survey of physical properties and their genetic controllers is an interesting story. I have some comments about the presentation and some concerns about the randomization-based null models (and hence the results downstream of these), but these may well be due to my own misunderstanding and I’d be very open to being corrected. The most important scientific comments are in bold as “randomisation method point 1 & 2” below.

For transparency, I am Iain Johnston and I am happy for this review to be treated as public domain. To my eyes my most important shortcomings as a reviewer are a lack of familiarity with the genetic players involved in the regulatory circuit under study.

Presentation comments

----

This is a huge paper and as such it is a challenge to read. That’s fine – readers should be challenged. But the challenge is compounded by the placement and styling of important information. With this many figures, the reader could be helped substantially more.

We are many times directed to multiple subfigures of di<erent figures to support a scientific point (e.g. “the size of the largest cells is moderately reduced in acr4-2 mutants (Figs. 1B, 4B, 4H–J, 5B, 5H–I, 216 K), and more greatly reduced in atml1-3 mutants (Figs. 1C, 4C, 4H–J, 5C, 5H–I, K).”, l215-6; also l222, 250, 255, 269, 272, 278, 298, perhaps some more). I’m sure I’m not alone in finding it challenging to make comparisons across multiple subfigures across multiple pages.

First – parts 4HIJ and 5HIK are the same data many times over and the best way of plotting these data is actually none of these options – they should be visualised as individual datapoints, with their profile expanded to reflect density (e.g. with ggbeeswarm), with the cell area scale on a logarithmic axis. This is also the case for 3D, 2E, 2F, and any other distributions of cell area in the SI.

Second – to my eyes the most important message from these comparisons is the similarities (or di<erences) in the impact of the di<erent mutants across the tissue types (sepal, early leaves, mature leaves). This could be done in one frame. Fig 5L is the best current example, but I’m confused about the normalisation (see below). Why not have pointwise, log-scaled, cell area distributions in a trellis plot for mutants (columns) and tissues (row)? Then the reader can scan across the trellis, see the similar patterns across tissues from the di<erent genetic perturbations, and get all the information

together.

Some suggestions for paring back content:

Fig 2GH don’t report much (anything?) more than their correlation coe<icients would

imply – could be removed

Fig 4A-G and Fig 5A-G could be combined into one “comparative microscopy” picture

(and 4H-K and 5H-L combined into a single frame as suggested above).

Fig 6 demonstrates the validity of the method – important, but not central to the main-

text narrative – could go to SI?

Fig 9B-C and Fig 10A,C could be combined into a single panel, then Figs 9 and 10 could

potentially be merged.

A connection should be made with https://journals.biologists.com/dev/article/144/23/4386/19257/Pavement-cells-and-

the-topology-puzzle

which, while not focusing specifically on giant cells, does consider generative mechanisms and development of size distributions in pavement cells and their connectivity statistics.

https://www.cambridge.org/core/journals/quantitative-plant-biology/article/using-

quantitative-methods-to-understand-leaf-epidermal-

development/26AFBE802A863313E7EDD814A477691B

also provides a good overview of existing quantitative approaches to the question.

Scientific comments

----

L154 – what functional form do the tails of these distributions have? Exponential decay? If they are instead heavier (e.g. power-law) it could speak to an interesting mechanism of generation

L161 – H2B-TFP fluorescence is used as a proxy for nuclear DNA ploidy. Why does it have such a continuous distribution (horizontal axis of Fig 2G-H)? Wouldn’t we expect to see somewhat discrete jumps corresponding to integer / 2^n values? If it’s because of noise in the readout, does this challenge the interpretation?

L187 and throughout – as above, please plot all cell areas on a logarithmic scale – they invariably spans many orders of magnitude and we lose a lot of info from their “clumping” at low values!

L187 (here and generally) – area is used synonymously with size. So is area a good readout of volume? Does it matter, in terms of generating/control mechanisms? It’s fine if this is all a theory of cell area as opposed to size (which usually means volume) – but I couldn’t find this mentioned explicitly.

L201 – “the large cells on the blade on one side frequently did not form directly opposite the large cells on the blade on the other side”. If this is to be a scientific result, it needs quantification with respect to a reasonable null hypothesis. From eyeballing Fig 3C it actually looks to me like there is some notable symmetry in the large cell spatial distributions – especially given the noise in the overall cell layout. The null hypothesis ”every large cell must be exactly opposite another one” is too precise to be interesting; what values of which test statistic would lead the authors to accept that there was symmetry? (NB, scientifically I believe the large cells are established independently – what mechanism could cross the leaf? – but statistically it’s not safe to use this as evidence in its current form)

L260 – This MDS plotting (methods l730-733). What is the deal with the normalisation? Every “replicate” is scaled to have average (mean?) 1 – is that every instance of every mutant line in every tissue type? What happens if you don’t do this? You’re losing a tremendous amount of information about cell size, which clearly varies a lot across cases. If the unnormalised version doesn’t show the same clustering structure, then the genetic dependence of shape is rather higher-order than the genetic dependence of size – which should be mentioned if so!

L340 – The randomisation method (point 1). This is probably my biggest concern with the paper from here on. For the pavement cells, the randomisation produces cell geometries that look very much more convex that the biology (see pic below). This is clear from the examples given across all the pavement cell sets in Figs 7 and 8. This will certainly a<ect the statistics of connectivity between cells, as sets of cells with less convoluted boundaries will have fewer opportunities to contact each other. For the sepal cells, the randomisation destroys linear structure in the tissue (seen in the computational example in Fig. 9C). This might also influence the connectivity statistics for giant cells, as it likely gives more opportunities for small cells to sneak in between them than is actually the case in the adjacent ranks seen in the real bio.

(Reviewer’s figure – see attachment)

Because of these rather dramatic di<erences in physical arrangement, I think the randomisation method risks introducing fairly substantial artefacts into the biological structure of the null models. My concern is that the e<ects reported as departures from the random null model are in the direction of what we’d expect from such artefacts – and as such it’s hard to say whether we’re seeing signals of a true biological e<ect, or a comparison with an imperfect null model.

As a reviewer I can’t drop a fairly strong criticism like this without proposing a possible solution. It is clearly impossible to pick up a bunch of jigsaw pieces, rearrange them, and put them back into a new and perfect tesselation that fills the original plane. But I wonder if a randomisation method that removes shared sections and introduces new division planes would come close. For example, in the image below, remove boundary between red points and add boundary between yellow points:

(Reviewer’s figure – see attachment)

This way you’re destroying some giant cells and creating others in arbitrary positions through the cell while preserving the key morphology of pavement cells. This would also of course work for the sepal cells, redistributing the giant cells while preserving the side-by-side linear rows. At the very least, if this gave similar results to the existing randomisation approach, it could be claimed that two different choices for randomisation agreed in their outputs.

L354 – “In addition, it was less probable to find isolated giant cells, and more probable to find giant cells in contact with two or more other giant cells compared with what was expected by chance (Fig. 7D).” – not really in addition, as once we know this statement we know that the mean must be higher.

Fig. 7C, 8B, 8H – we can’t have p = 0 here, only p < 0.0025.

L385 – “regulated by a self-catalytic feedback loop” – but this isn’t shown in the figure (9A)?

L387 – some details of how cell contents are inherited at divisions, and how spatial structure emerges in the model (do cells push each other out of the way?) is essential for interpretation here (see comments on Methods).

L414 – first – what is the di<erence between Fig 9D and Fig 10B (right)? Don’t they both show giant neighbour counts in simulated tissues at the final time point? Why are they different?

Second – so, if I understand, the argument isn’t that giant cells acquire more neighbours over time (indeed Fig. 10A shows the giant interactions are preserved and the red bars in Figs. 10B and D look like they have the same values) – but that the randomised null model gives progressively lower giant cell neighbour counts as time progresses. But doesn’t this mean (randomisation method point 2):

- that giant cells are positioned randomly in space – because that’s how they simulations start

- that they remain positioned randomly in space during development – because the statistics of their neighbouring interactions with other giant cells (irrespective of any null model) don’t change

- that randomly swapping cell identities becomes a worse null model as the tissue develops, leading to giant cells departing more from its predictions? This is what we’d expect, because the developmental history of a cell in a tissue influences its physical situation, and so the extent to which a cell is not just a random independent object grows with its history.

I think a fundamental point here is that “agreement with a null model that randomises cells” is taken as synonymous with “randomly arranged”, and “departure from a null model that randomises cells” is taken as synonymous with “clustered”. But actually the degree of clustering – the connectivity of the giant cells – doesn’t change, and nor does their arrangement within the tissue (as seen in simulation in Fig 10A and in real bio in Fig 10C). So it can’t be meaningful to say “more clustered” over time (l403, 407, 426, 455) – the change is in the null model predictions, not the direct observations.

It feels like a “quenching” picture, to borrow from physics – interactions between giant cells are quenched while the entropy of the rest of the tissue increases through the constant churn of cell divisions. They’re not getting more clustered, but the order they remember from the simpler conditions gets more and more different with respect to the increasing entropy of the tissue as a whole. This is still an interesting message, but I suggest the phrasing be changed throughout to reflect the static nature of the giant cell statistics and the dynamic nature of everything else.

Methods

----

I couldn’t find a link to the software implementation.

For NetworkX, matplotlib, seaborn, and any other libraries involved, please cite the appropriate references to credit the authors in a trackable way rather than just giving a URL.

I think many people who otherwise find the paper interesting will be put off by how the math model is presented on l876-894. It assumes a lot of familiarity with how (stochastic) systems biology models are constructed. Why not have a sentence or two describing what these complicated-looking expressions are actually doing? E.g. “the terms on the right hand side of Eqn 1 describe constitutive expression, self-activation (using an enzyme-kinetic model), degradation, and a (scaled) random term modelling thermodynamic fluctuations”. NB – the inline expression on l893 is illegible in my review copy but I’m assuming it’s describing zero-autocorrelation unit noise.

Where do the parameter values come from, and how do we know they’re reasonable?

What are the inheritance rules for ATML1 and Target? i.e. when a cell divides, are the levels of these variables reset, inherited proportional to daughter cell volume, inherited equally, …? This would seem to be essential in considering spatial correlations in cell behaviour across a growing tissue.

How does spatial structure emerge? Looking at Fig 9B I can see some cells that have divided with a vertical (in the figure) division plane. What’s to stop this occurring more than once and pushing apart the lateral structure of the tissue? Why do the giant cells (and others) have thickness differences (tapering)? It seems like there’s a tremendous amount of detail here that is swept into “the multicellular tissue grows anisotropically” (L910).

---

## [Decision Letter · Decision Letter 2]

20 Aug 2025

Dear Dr Roeder,

My name is Luke Smith - I am an editor at PLOS Biology and I am writing on behalf of my colleague, Dr. Ines Alvarez-Garcia, who is out of the office at the moment. Thank you for your patience while we assessed your revised PLOS Biology manuscript "A common pathway controls cell size in the sepal and leaf epidermis leading to a non-random pattern of giant cells". Your manuscript has now been evaluated by the PLOS Biology editors, the Academic Editor and by the original reviewer 2.

The reviews are appended below, and you will see that reviewer 2 thinks the study has been greatly strengthened in this revision. However, Reviewer 2 has a number of last suggestions to improve the study further, largely by refining the precision of the claims made here and by providing more nuanced discussion of the results.

In light of the reviews, we are pleased to offer you the opportunity to address these remaining points in a revision that we anticipate should not take you very long. We will then assess your revised manuscript and your response to the reviewers' comments with our Academic Editor aiming to avoid further rounds of peer-review, although we might need to consult with the reviewers, depending on the nature of the revisions.

**As you address reviewer 2's comments, we ask that you please also:

1) Update your financial disclosure statement, in our editorial manager system to include the full name of each funder and the URL of their website.

2) Add a sentence to each figure legend pointing readers to the underlying data, which you have deposited on OSF. For example, you can add the sentence "The underlying data for this figure can be found at pen Science Framework (osf.io), DOI:10.17605/OSF.IO/RFCWS"

**IMPORTANT - SUBMITTING YOUR REVISION**

*Resubmission Checklist*

*Published Peer Review*

*PLOS Data Policy*

*Blot and Gel Data Policy*

Sincerely,

Luke

Lucas Smith, PhD

Senior Editor

PLOS Biology

lsmith@plos.org

--on behalf of--

Ines Alvarez-Garcia, PhD

Senior Editor

PLOS Biology

REVIEWS:

Reviewer #2, Iain Johnston (note, reviewer 2 has signed this review): The authors have input a tremendous amount of work into their edits and to my eyes things are substantially improved. My main points about the difficult layout are resolved, lots of individual questions have been answered, and I really appreciate the additional exploration and alternative approach in the randomisation questions I had earlier. The technical content (esp Figs. S18-19) has resolved my questions about the computational randomisation; the rhetorical changes have largely addressed my points about the interpretation, although it's on that point that I still have (minor) points to raise, as well as a few easy fixes below.

The authors have rephrased arguments about clustering; to respond to their question in the response letter I think "more clustered with respect to the rest of the tissue" would be a reasonable phrasing.

There remain in the manuscript a lot of mentions of "more than expected by chance". I think this discussion has highlighted the need to be more precise about this. "By chance" could mean "if every aspect of development was completely random" or "if cell identities in the final tissue were randomly assigned". The latter is the real result, but without reading the full paper I imagine most readers' instinct will be more along the lines of the former. I think that throughout (including l28 in the abstract), mentions of "by chance" should be replaced with a more precise statement e.g. "than expected from a randomised null model".

There also remain some implications that a "non-random mechanism" is responsible for the observed patterning. But I would question this (as I previously did) -- the "mechanism" is not directed, deterministic control, but rather an expansion of randomness in cells that aren't giant. It is not a "non-random mechanism", but a difference in the extent of randomness in giant vs other cells that is shaping the arrangement. I expand on this labelled ** below.

IJ

l120 -- "more than expected by chance" -> "more than expected by chance, reflecting the tissue's developmental history".

l133 puzzle -> puzzle piece [the puzzle is usually a rectangle!]

l188 -- this continuous value for total nuclear signal still feels strange. My question was less about whether signal is genuinely correlated with ploidy, and more about whether the obvious noise in the signal (any departure from tight integer peaks is a departure from perfect readout of ploidy) challenges the interpretation. Perhaps not -- but I think it would feel more comfortable if the continuous and noisy nature of this readout was explicitly mentioned.

l217 -- at the same proximal-distal position -> occupy the same proximal-distal arrangements [we're talking about the set of cells, right?]

l280 -- no difference -> no statistically significant difference

l282 -- this result feels awkwardly like cherry picking -- just for this mutant, one statistic (number per area) didn't give p < 0.05, so we look at another (fractional area of giants) which does. We're not normally allowed to do this, because there'll always be some statistic that gives p < 0.05 under the null hypothesis. Did the alternative statistic (fractional area of giants) also give p < 0.05 in the other mutants?

l291 -- the entire cell-size distribution -> other aspects of the cell-size distribution

l296 -- cell size distributions -> normalised cell size distributions

[note that there's some inconsistency in the use of "cell size" vs "cell-size"]

l333 -- double space after having

l394 -- I'd remove "slightly" -- I think they are dramatically affected! -- but best to just objectively report "affected". I'd pull the science bit of this paragraph "Several measures... formed a null distribution" into a new paragraph below the rest of this one, separating the model setup/validation and the scientific objective.

l1514 -- also remove "slightly"

l401, l422, l426, l575 -- "than expected by chance" -> "than in a randomized null model"

**

l431 -- a non-random mechanism -- this is really my main question about the phrasing of the intepretation (as before). Is an expansion of external randomness around a quenched structure really a non-random mechanism? I don't think so -- to me a "non-random mechanism" implies some directed process, rather than quenched structure on a background of increasing entropy. It's actually explicitly the *expansion* of randomness that is inducing the signal of clustering. Perhaps it's safer to say "the giant cell arrangement must be subject to less random influence than the rest of the tissue" -- which is both true and reflects the actual mechanism that is revealed latter.

l454 -- "regulated biological mechanism" -- I think this is even more controversial. The preceding aspects of the paper have shown genetic regulation, but these simulation studies don't give direct support to the regulated nature of this "mechanism" governing arrangement. I would rephrase like the previous point, or just omit this -- the previous sections have already shown where regulation takes place.

l495 -- I'm not sure what the link to Fig. 7A should point to, but I don't think it should point to Fig. 7A

l502 -- same with link to Fig. 4D?

l550 -- "by chance" -- "under a random model where all final cells have random identities" -- or similar

l863 -- "such that the variances" -- but that's not the full story right? If you wanted to compare the variances you could just look at the absolute difference in variances as your MDS distance. Using Wasserstein means that you're also reporting all higher moments of the distributions too. That's fine, but this text shouldn't imply otherwise -- perhaps "such that all higher moments of the area distributions can be compared"

para l957 -- lots of Fig. 19 instead of Fig. S19

---

## [Editor Report · Decision Letter 3]

16 Oct 2025

Dear Dr Roeder,

Thank you for the submission of your revised Research Article entitled "A common pathway controls cell size in the sepal and leaf epidermis leading to a non-random pattern of giant cells" for publication in PLOS Biology. On behalf of my colleagues and the Academic Editor, Mark Estelle, I am delighted to let you know that we can in principle accept your manuscript for publication, provided you address any remaining formatting and reporting issues. These will be detailed in an email you should receive within 2-3 business days from our colleagues in the journal operations team; no action is required from you until then. Please note that we will not be able to formally accept your manuscript and schedule it for publication until you have completed any requested changes.

PRESS

Sincerely, 

Ines

--

Ines Alvarez-Garcia, PhD

Senior Editor

PLOS Biology
